

# The UBC ATMOX chamber: An 8 m3 LED-powered modular environmental chamber for indoor and outdoor atmospheric chemistry

Rickey J. M. Lee[1], Ayomide A. Akande[1], Saeid Kamal[2], Paul A. Heine[1], Pritesh Padhiar[3], David Tonkin[4], Wesley Rusinoff[4], Mohamad Rezaei[3,4], and Nadine Borduas-Dedekind[1]

[1]Department of Chemistry, University of British Columbia, Vancouver, V6T 1Z1, Canada
[2]LASIR, Department of Chemistry, University of British Columbia, Vancouver, V6T 1Z1, Canada
[3]Mechanical Engineering Services, Department of Chemistry, University of British Columbia, Vancouver, V6T 1Z1, Canada
[4]Electronic Engineering Services, Department of Chemistry, University of British Columbia, Vancouver, V6T 1Z1, Canada

**Correspondence:** Nadine Borduas-Dedekind (borduas@chem.ubc.ca)

**Abstract.** Environmental chambers are controlled reaction vessels used to investigate atmospheric processes such as photochemical smog, atmospheric fate of molecules and secondary organic aerosol formation. Environmental chambers are typically equipped with UV-A fluorescent lights with wavelengths between 350 and 410 nm or xenon lamps with wavelengths between 300 and 800 nm. However, these types of lights increase the temperature of the chamber, are energy intensive and are not tun-
5 able to specific wavelengths. Fluorescent lights are also becoming redundant in the light industry. To address these issues, we prototyped the use of light-emitting diode (LED) lights from Violumas on our 8 m$^3$ environmental chamber for photochemical experiments to enable stratospheric, tropospheric and indoor light conditions. The University of British Columbia (UBC)'s Advanced Techniques for Mechanisms of OXidation (ATMOX) chamber was assembled with custom-made wide-angle LEDs of six different wavelengths from Violumas: 275, 310, 325, 340, 365, 385 nm. We also added LED grow plant lights (Feit
10 Electric) for irradiance between 450 and 630 nm. The LEDs were wired to a potentiometer control panel to modulate their output on a per wavelength basis. We used a total of 1440 custom LEDs and 1320 commercial grow plant LEDs, costing USD\$ 44,951 and USD\$ 1,300, respectively. We demonstrate their energy efficiency, their ability to generate less heat, and their ability to generate wavelength-specific photochemical processes. Furthermore, chemical actinometry using NO$_2$ enabled us to calculate a photolysis rate (J$_{NOx}$) ranging from $2.28 \times 10^{-4}$ to $4.93 \times 10^{-3}$ s$^{-1}$, which is nicely comparable to $4.50 \times$
15 $10^{-3}$ s$^{-1}$ in Vancouver, Canada during the summer solstice.

  In addition to the lights, the UBC ATMOX chamber was designed to be particularly modular. The chamber frame has 12 aluminum T-slot rails ($2.66 \times 2.66 \times 3$ m, 80/20), and a pulley system to enable the 8 m$^3$ bag to collapse and inflate, to perform batch or continuous mode experiments. The Teflon chamber bag has sealable openings at each corner to allow access to the interior of the bad for regular thorough cleaning. Overall, our chamber is allowing us to study topics of current interest in
20 atmospheric chemistry: from the fate of indoor air fragrances to cannabis emissions and from wildfire aerosol photochemical changes to the biogeochemical cycling of selenium.



**Short summary:**The University of British Columbia (UBC)'s Advanced Techniques for Mechanisms of OXidation (AT-MOX) chamber is a modular 8 m$^3$ environmental chamber capable of operating under batch and continuous mode experiments with a unique setup of light-emitting diodes (LEDs) producing irradiance peaks at 275, 310, 325, 340, 365, 385 nm, as well as between 450 and 630 nm. This chamber enables wavelength-specific photochemical experiments without temperature increases while being energy efficient.

# 1 Introduction

## 1.1 Overview of environmental chamber characteristics

The atmospheric oxidation of volatile organic compounds (VOCs) leads to the formation of oxidation products which can form pollutants such as ozone and secondary organic aerosols (SOA) (Ylisirniö et al., 2020; Yang et al., 2024). These pollutants impact air quality through human exposure, and climate through new particle formation (Zhang et al., 2015; Shiraiwa et al., 2017; Lei et al., 2024). Studying these processes in the natural environment is challenging due to the complex matrix of atmospheric pollutants, which undergo chemical production and loss processes but also emission, transportation, and deposition (Zhang et al., 2015). Additionally, ambient measurements are inherently limited because temperature, relative humidity, light, and chemical composition cannot be independently controlled (Jorga et al., 2020). Balancing research between field observations, laboratory experiments, and modeling studies strengthen our understanding of chemical processes (Abbatt et al., 2014; Burkholder et al., 2017).

To address the complexity of the transformations of gas phase and particulate phase molecules in ambient air, researchers use environmental chambers to isolate and study specific chemical reactions under controlled atmospheric conditions (Chu et al., 2022; Hidy, 2019; Bruns et al., 2015). These chambers enable the control of light, temperature, and relative humidity, as well as which compounds are introduced and generated at which time. Control of these factors allows for the targeted analysis of atmospherically relevant chemical reactions, such as oxidation of VOCs, under controlled conditions without interference from other pollutants, temperature fluctuations, or variations in relative humidity (Hidy, 2019; Kenagy et al., 2024; Atkinson, 2007; Wennberg et al., 2018). (Nguyen et al., 2023) have recently reported on a data repository for these simulation chamber parameters. These chamber experiments are then used in modeling studies, improving our predictive capabilities for atmospheric chemistry. Drawing on GEOS-Chem and box-model simulations (F0AM), Kenagy et al. (2024) illustrated how factors such as reactant concentrations, photolysis rates, and reaction timescales shape organic peroxy radical RO$_2$ pathways. By mapping these parameters onto chamber setups, researchers can decide how to scale their chamber size, light intensity and oxidation concentration, depending on the specific reactions under study. This scaling ensures that longer-lived or more complex chemicals have enough time and the right conditions to undergo repeated oxidation steps in a realistic lab setting.

The chamber material of the reaction vessel is an important consideration to minimize sorption and reaction losses to the chamber walls. Fluorinated polymers like Teflon (PTFE) are commonly used (Morris et al., 2024; Shao et al., 2022; Deming et al., 2019; McMurry and Grosjean, 1985; Krechmer et al., 2017, 2020; Bilsback et al., 2023), and quartz has also proven to be an excellent non-reactive surface (Ma et al., 2022). Chamber bags can also be hung within cooling chambers to access a



wide-range of temperatures (Paulsen et al., 2005; Bates et al., 2021; Boyd et al., 2015). To study cloud formation, access to
a range of pressures is required and therefore researchers have used aluminum (AIDA)(Vallon et al., 2022) and stainless steel
(PINE)(Möhler et al., 2021) for expansion chambers. Overall, choosing the right vessel material depends on the research needs.

Environmental chambers can be employed under batch or continuous flow modes (Krechmer et al., 2020). In batch mode,
the volume of air changes without dilution. Practically, the reaction vessel must be able to collapse as the volume decreases as
a function of time. To manage a changing volume, the fluoropolymer chamber is mounted on movable frames or rail systems
that accommodate expansion and contraction of the reaction vessel (Paulsen et al., 2005). For example, a fluoropolymer bag
is filled with air and the target compound which is then oxidized as a function of time and monitored by instruments pulling
the air inside the chamber (Krechmer et al., 2020). This experiment is useful for chemical kinetics and mechanistic studies, as
there are no dilutions to take into account but, it has the drawback of increasing the surface area to volume ratio as a function
of time. This drawback affects wall losses for less volatile compounds (Deming et al., 2019; Krechmer et al., 2020; Leskinen
et al., 2015; Nakagawa et al., 2024; Wang et al., 2018; Zong et al., 2023; Nguyen et al., 2023). By contrast, in continuous
flow mode, the chamber volume remains constant while air and reactants are continuously supplied, creating more realistic
steady conditions for extended periods if the reaction timescale is relatively short and does not exceed the flow residence time
(Zhang et al., 2018; Krechmer et al., 2020). For example, secondary organic aerosol (SOA) production is often conducted under
continuous mode to produce material to be collected onto a filter (Mahrt et al., 2022). The choice of chamber operation mode
depends on the research question, and thus a modular chamber, like the one described here, can have access to both abilities.

## 1.2  Examples of specific environmental chambers

Within the available environmental chambers in atmospheric chemistry and physics research, we consulted and considered the
designs and capabilities of chambers listed in Table 1 and beyond. This list is not exhaustive in any way, but represents the
breadth of designs (outdoor vs indoor, light sources, size and characteristics) and operations that inspired our own chamber
development at the University of British Columbia.

A wide range of environmental chambers have been developed globally, each tailored to specific research objectives. Outdoor
chambers use sunlight to investigate photochemical degradation and oxidation processes under ambient conditions, allowing
direct measurements of radical species like OH, $HO_2$, and $RO_2$ under near-real atmospheric scenarios, such as the 280-370 $m^3$
double-layer FEP SAPHIR chamber in Germany (Rohrer et al., 2005; Cho et al., 2023), and the hemispherical 200 $m^3$ FEP
Teflon EUPHORE chambers in Spain (Zádor et al., 2006). The 100 $m^3$ double layer FEP chamber at the Chinese Academy
of Sciences (AESS-RCEES in Beijing) likewise uses sunlight and can draw in either zero air or ambient urban air with a
specialized intake system, facilitating smog simulations (Ren et al., 2024). Indoor environmental chambers can range in size
from 1 $m^3$ to 10s of $m^3$, and can have a number of controllable parameters like temperature, pressure and relative humidity.
For example, the CERN CLOUD chamber is a 26.1 $m^3$ stainless steel cylinder to study new particle formation with access to
the CERN proton synchrotron (Kirkby et al., 2011, 2023). Furthermore, Kaltsonoudis et al. (2019) opted for a dual-chamber
setup, using one chamber as a baseline and the second chamber as a perturbation chamber. Caltech employs dual 28 $m^3$ FEP
reactors for smog and SOA studies (Cocker et al., 2001) while Georgia Tech uses dual 12 $m^3$ Teflon bags operated in a climate-





controlled enclosure adjustable to 4-40°C, which has been applied to cellular oxidative stress (Ng et al., 2019). At MIT, a
7.5 m³ Teflon chamber has been characterized for multi-generational VOC oxidation experiments (Isaacman-VanWertz et al.,
2018). UC Davis operates a 10 m³ Teflon environmental chamber with UV blacklights and an adjustable temperature range (-20
to 60°C) to study hydrocarbon oxidation and aerosol formation across a wide span of NOx levels, humidity, and seed aerosol
conditions (Nakagawa et al., 2024; Bates et al., 2021). These chambers demonstrate how chamber size, materials, operation
mode, and light sources are deliberately chosen to support each facility's specific atmospheric research objectives, from urban
smog chemistry to climate-related aerosol–cloud interactions.

## 1.3   Lighting for environmental chambers

A variety of environmental chambers have been designed to use sunlight for photochemistry, such as the chambers at UPatras
(Kaltsonoudis et al., 2019), SAPHIR (Rohrer et al., 2005), HELIOS (Ren et al., 2017) and EUPHORE (Zádor et al., 2006),
because of favorable weather and high solar irradiance. However, this approach is impractical in regions with frequent cloud
cover (e.g. Vancouver) (Seakins, 2010). Consequently, indoor environmental chambers rely on advanced artificial lighting
systems to drive photochemical experiments (Kim et al., 2024). Common setups have traditionally used xenon arc lamps or
arrays of fluorescent bulbs as light sources (Kim et al., 2024; Paulsen et al., 2005; Leskinen et al., 2015; Kaltsonoudis et al.,
2019), but we wanted to explore light-emitting diodes (LEDs) for our source of lighting.

Fluorescent lights are cheap to fabricate, but are gradually being replaced by LEDs due to technological advancements,
including energy efficiency (Ganandran et al., 2014; Bhattarai et al., 2024). Indeed, LEDs can deliver the same illumina-
tion as fluorescent lights while using roughly 30-40% less power (Ganandran et al., 2014). Furthermore, LEDs do not use
mercury-based phosphors integral to fluorescent bulbs, making them safer and avoiding the disposal of hazardous material
(Morgan Pattison et al., 2018). Consequently, research groups are moving away from fluorescent fixtures in favor of more
efficient LED arrays (Vallon et al., 2022; Wu et al., 2021; Chu et al., 2022).

In the context of atmospheric photochemistry, LEDs have a major advantage over fluorescent lighting; LEDs emit a narrow
range of wavelengths, enabling wavelength-specific photochemistry (Muramoto et al., 2014; Bhattarai et al., 2024). Arrays of
LEDs can replicate the solar spectrum (Sun et al., 2022). Most recently, the Aerosol Interactions and Dynamics in the Atmo-
sphere (AIDA) chamber at the Karlsruhe Institute of Technology in Germany is a large 84.5 m³ cylindrical aluminum vessel
outfitted with an LED-based solar simulator consisting of LED banks arranged around the top of the chamber (Vallon et al.,
2022). This system produced a combined light spectrum that roughly covers 300-530 nm, effectively mimicking tropospheric
sunlight in the UV–visible range (Vallon et al., 2022). Following this advantage, our chamber also incorporates LED lighting
to capitalize on the precise wavelength tunability that LEDs provide to produce our own modular solar simulator.

To simulate stratospheric, tropospheric, and indoor light conditions in one environmental chamber, we developed the 8
m³ Advanced Techniques for Mechanisms of OXidation (ATMOX) chamber at the University of British Columbia (UBC).
Here, we detail the chamber's construction, the rationale behind each design choice, and its effectiveness in simulating a
range of photochemical conditions using custom-designed LEDs. It is also equipped with a rail suspension that allows for





contraction and expansion, providing a modular framework adaptable to batch and continuous modes, while sealed ports ensure straightforward cleaning and maintenance.



**Table 1.** List of environmental chambers and their key characteristics that have been used for various types of atmospheric chemistry around the world

| Location | Light type | Light supplier | Volume (m³) | Material | Characteristics | Reference |
|---|---|---|---|---|---|---|
| UBC, Canada | LEDs | Violumas | 8 | PFA | Tunable photochemistry Precise T, RH control | This study |
| PKU, China | Fluorescent | 40 W, Bulb-T12, GE, USA | 2 | FEP | Seed generation system | Zong et al. (2023) |
| JNU, China | Undisclosed | Undisclosed | 8 | FEP | Vehicle mounted dual chamber | Wang et al. (2023) |
| KIT, Germany | LEDs | Various LEDs from LG Innotek, EPIGAP, Seoul Viosys, Osram | 84.5 | Aluminum | Precise p, T, RH control | Vallon et al. (2022) |
| BUCT, China | Fluorescent | 60 W Philips/ 10R PL, Germany | 10 | Quartz | Quartz wall material for minimal wall loss | Ma et al. (2022) |
| CU Boulder, USA | Fluorescent | Undisclosed | 20 | Teflon | Gas-particle partitioning experiments | Liu et al. (2019); Krechmer et al. (2017) |
| GT, USA | Fluorescent | Sylvania, 24922 | 12 | Teflon | Dual smog chamber | Ng et al. (2019); Boyd et al. (2015) |
| UPatras, Greece | Fluorescent | Osram, L36W/73 | 1.5 | FEP | Portable dual smog chamber | Kaltsonoudis et al. (2019) |
| NC A&T, USA | Fluorescent | F30T8/350BL/ECO, Sylvania | 9 | FEP | Combustion gas-phase chamber experiments | Smith et al. (2019) |
| ICARE-CNRS, France | Outdoor, Xenon | Outdoor chamber Sylvania F 36 W/T8 | 90 | FEP | Outdoor and xenon light source, large V | Ren et al. (2017) |
| UEF, Finland | Fluorescent | BLB and Sylvania F 40 W/350 BL | 29 | FEP | Combustion aerosol chamber experiments | Leskinen et al. (2015) |
| MIT, USA | Fluorescent | Sylvania BL350 eco | 7.5 | PFA | T controlled room | Hunter et al. (2014) |
| PSI, Switzerland | Fluorescent | Cleo Performance solarium lamps, Phillips | 9 | FEP | Mobile environmental reaction chamber | Platt et al. (2013) |
| Jülich, Germany | Outdoor | Outdoor chamber | 270 | FEP | Outdoor chamber | Rohrer et al. (2005) |
| PSI, Switzerland | Xenon | XBO 4000 W/HS, OSRAM | 27 | FEP | Simulate particle formation and growth | Paulsen et al. (2005) |
| CalTech, USA | Fluorescent | Sylvania 350BL | 28 | FEP | Dual smog chamber | Cocker et al. (2001) |





## 2 Methods

### 2.1 Chamber experimental setup

#### 2.1.1 Chamber design

The UBC ATMOX chamber's exterior frame was constructed by cutting 80/20 aluminum bars (1.5" × 3" lite t-slotted extrusion 145") into 2.5 m bars for the base, and 3 m bars for the height (Figure 1a). Inside corner brackets (15 S 2 hole) were used to connect the 80/20 aluminum bars into a cubic frame. These brackets were screwed (5/16-18 × 11/16" flanged button head socket cap screw (BHSCS)) into threading (15 S econ t-nut 5/16-18 thread) placed inside the t-shaped grooves called t-slots of the 80/20 aluminum bars.

The PFA bag (Ingeniven, .005" thick) was mounted onto metal hook rollers which slotted into the T-slots of the 80/20 aluminum bar to form an 80/20 aluminum railing (Figures 1b, S1, S2). Stainless steel rods were inserted into the belt loops of the bag (Figure 1b). The bag was then lifted and hooked onto chains hanging onto metal hook rollers. The volume of the bag is expandable, so the pulley can be used to contract the bag in a purging experiment, for example. This ability enables modularity on the volume of the chamber as well as batch mode experiments and cleaning procedures.

Blackout blind material (30 m by 1 m) was purchased from Fabricana and sewn with a loop (Figure S3). An aluminum rod was inserted into this loop and attached to metal standoffs screwed into the T-slot extrusion (Figure 1c). The curtains are necessary to prevent indoor light from contributing to the photochemistry inside the bag, and to protect people working in the laboratory from the 275 nm LEDs.



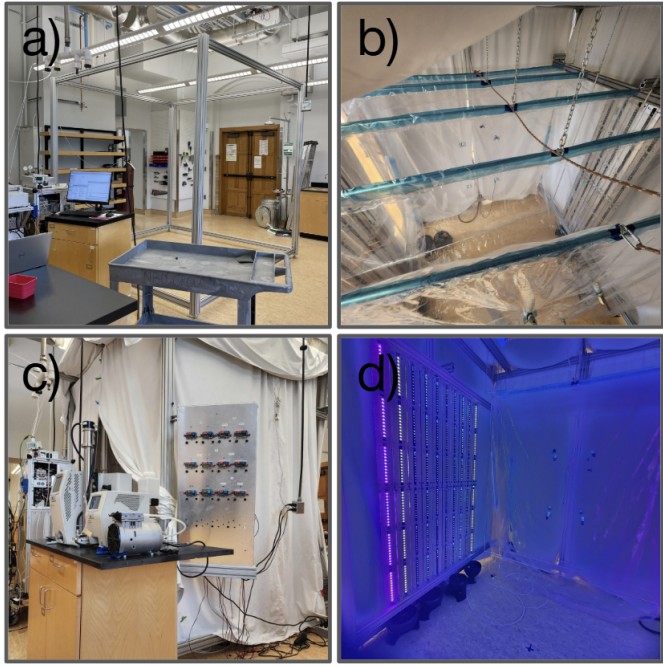

**Figure 1.** Photographs of the Advanced Techniques for Mechanisms of OXidation (ATMOX) environmental chamber at UBC during four stages of assembly and operation: (a) The 80/20 aluminum frame that forms the modular skeleton of the chamber. (b) The overhead pulley system used to expand and compress the chamber PFA bag. (c) The chamber in operation, with instruments such as a scanning mobility particle sizer (SMPS) and a Vocus PTR-ToF-MS sampling directly from the chamber, as well as the control panel for the LEDs. (d) The interior of the chamber illuminated by the multi-wavelength Violumas LED arrays on the side panels and the LED grow lights from the top of the chamber. See Figure 3 for a sketch of the ATMOX chamber.

### 2.1.2 Instrumentation

The chamber was equipped with instrumentation capable of measuring both gas and particle phase species to study their transformations (see Figures 1c and 3). For monitoring particle phase processes, the scanning mobility particle sizer (SMPS) (Model 3082. TSI Inc.) was used to measure ultrafine particles from 2 - 1000 nm. For gas-phase processes, the Vocus PTR-ToF-MS (2R, Aerodyne/Tofwerk) measured VOCs, (Krechmer et al., 2018), the Thermo Fisher 49i and 42i measured $NO_2$ and NO, and $O_3$, respectively. These instrumentation were used for studies in our laboratory including the ozonolysis of cashmeran (Akande and Borduas-Dedekind, 2025) and the photolysis of organo-selenide compounds (Heine and Borduas-Dedekind, 2023; Heine et al., 2025).

We had additional instrumentation to keep track of standard parameters in the chamber including probes and spectrometers. A relative humidity, temperature, and pressure (RHTP) sensor wrapped in polytetrafluoroethylene (PTFE) tape monitored the temperature of the chamber. For the spectroscopic measurements, we used the Ocean Optics FLAME-T-UV-VIS spectrophotometer (QP600-1-XSR fiber optic and CC-3-UV-S cosine receptor). Spectroscopic measurements were taken at the chamber's



center, with the cosine receptor directed towards the north and south facing LEDs, to characterize the photochemical environment experienced by aerosols at the center of the bag from all directions as the cosine receptor captures light in a 180 degree

angle (Figure S4). These measurements were summed to get the overall spectrum. During expansion and contraction of the bag, we also used an oil-less pump (617CA22 Thomas) to remove air and compress the bag when not in use.

### 2.1.3 Chamber cleaning procedure

The chamber was cleaned by rinsing the bag using a commercial pressure washer containing 18.2 MΩ water generated using

a PURELAB Option Q-7 system. The bag was flushed with particle-free air overnight. To ensure the chamber was clean, the SMPS and NOx analyzer sampled the chamber to ensure the chamber was particle-free and had minimal NOx.

## 2.2 Light emitting diodes (LEDs)

### 2.2.1 Choice and purchase of LEDs

We purchased LEDs from two suppliers: Violumas and FEIT Electric. We purchased 12 Violumas LEDs pre-installed on a

printed circuit board (PCB) board. The custom Violumas LEDs were built with a 60 degree viewing lens (Moreno and Viveros-Méndez, 2021), to increase the diffusivity of the light. These LEDs use a 3-pad structure to allow rapid thermal dissipation through the pillar structure of the LED chip and metal core printed circuit board (MCPCB). This innovation minimizes thermal resistance, thus increasing the optical output and the LED lifetime (Fredes et al., 2022).

The LED industry is rapidly evolving and new wavelengths are being added to the market regularly. For example, our

first purchase in December 2021 was for 310 nm and 365 nm LEDs (USD$759.70), but by January 2024, 325 and 340 nm LEDs came onto the market (Table 2). In total, we placed four orders of LEDs throughout December 2021 to September 2024 with a total cost of approximately 52,000 USD. (March 2023: USD$2,080.08; July 2023: USD$10,660.41; January 2024: USD$18,630.00 ; and July 2024: USD$9,963.00) (Table 2). The cost of the heatsinks was approximately 14.5% of the cost of the LEDs.





**Table 2.** LEDs and heatsinks were purchased from Violumas between December 2021 and September 2024 with the listed unit costs. All totals are shown before taxes. The UBC ATMOX chamber has 1440 Violumas LEDs and 1320 LEDs from the plant grow lights.

| Component | Year(s) purchased | Total quantity | Approximate total cost (USD$) |
|---|---|---|---|
| 275 nm LED | 2023 | 72 | 2,700 |
| 310 nm LED | 2023 | 288 | 11,300 |
| 325 nm LED | 2024 | 324 | 14,580 |
| 340 nm LED | 2024 | 324 | 13,365 |
| 365 nm LED | 2022, 2023 | 288 | 2,034 |
| 385 nm LED | 2022, 2023 | 144 | 972 |
| Grow light LEDs | 2023, 2024 | 1320 | 1,300 |
| Total for LEDs | | 2760 | 46,251 |
| Heatsinks | 2022, 2023, 2024 | 55 | 6,545 |

### 2.2.2 LED current and voltage

The LED array in this study included six distinct LED models from Violumas, each equipped with a 60° fused silica lens. These LEDs covered wavelengths of 275 nm, 310 nm, 325 nm, 340 nm, 365 nm, and 385 nm, with each wavelength chosen for targeted photochemical experiments. The UV-C LED (VC12X1C48L6-275-V1) operated at a forward current of 1400 mA, producing a typical radiant flux of 1.32 W at a forward voltage of approximately 37.2 V. Similarly, the UV-B LED (VC12X1C48L6-310-V1) also operated at 1400 mA, providing a radiant flux of about 1.44 W at a forward voltage of 36.0 V. The UV-A LED models WC12X1C40L6-325 and WC12X1C40L6-340-V1 ran at a forward current of 700 mA, with typical radiant fluxes of 660 mW and 1900 mW, respectively. Their corresponding forward voltages were approximately 28.5 V (325 nm) and 27.0 V (340 nm). Finally, VC12X1C45L6-365 and VC12X1C45L6-385 also operated at 700 mA, emitting significantly higher radiant fluxes of approximately 6 W (365 nm) and 13 W (385 nm). Their typical forward voltages were 49.2 V (365 nm) and 44.8 V (385 nm). All LEDs featured integrated thermal management technology, built-in ESD protection, and solder-free connectors, facilitating efficient assembly and operation.

### 2.2.3 Assembly of the LED lights, bars and panels

All assembly took place on an electrostatic discharge (ESD)-protected bench with a wrist strap to safeguard the LED strips. Each strip was mounted onto a custom-cut aluminum panel, separated by Kapton tape to prevent static buildup. A two-circuit terminal block was attached first, followed by inserting rubber grommets into the wire-passage holes. After checking the strip's Kapton tape and replacing any torn sections, the strip was placed on the factory-applied thermal grease and fastened with plastic screws until a slight ooze signaled proper contact. Black electrical tape covered the screws to shield them from UV-aging. Three strips were oriented the same way and wired in series (Figure S5) with 20 AWG, 600 V-rated wire, routing the connections behind the panel through grommeted holes and securing them with Kapton. The LED panels were mounted perpendicular to





the sampling ports to preserve unobstructed access during experiments. Finally, M8 screws mounted the panel onto the frame. We found that the bag material was transparent to 94% of the LED output, and thus light transmission through the chamber was not a concern (Figure S6).

### 2.2.4   LED Wiring

The LED bars connected to a power supply (Meanwell ELG/HLG series, type B, Figure 2c) included a current control potentiometer wired to the driver. A current display module (MAX472 board) wired in series with the LEDs reads the output of the current at all times (Figure 2a). The wiring diagram (Figure S7) of a single power supply and a single display module is shown below (Figure 2c, 2d). The same circuit was used for all power supplies.

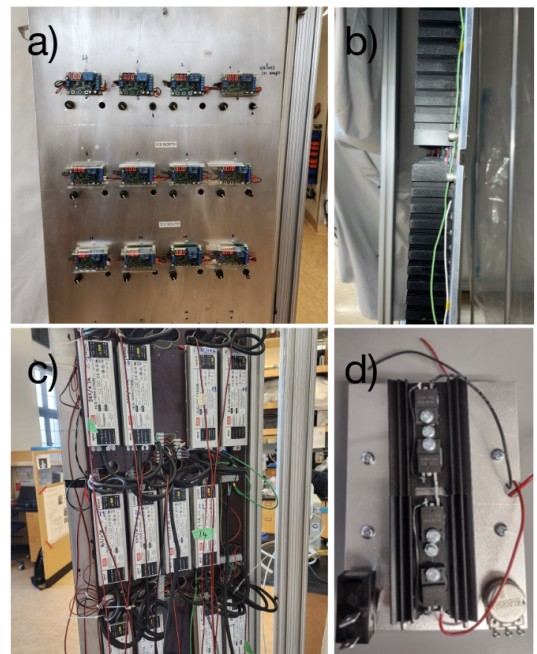

**Figure 2.** The electrical control panel for the LEDs are shown in the images. (a) A user-accessible control panel reads out the current driven through each LED. Each current display module is linked to a panel of 3 bars consisting of 12 LEDs each. The black switch turns the LEDs on and off, while the knobs adjusts the current. The LEDs are made of three components: (b) The heatsinks are connected to the LED bar via an aluminum mounting frame (side view). (c) Power supply units are mounted at the rear of the panel shown in (a). (d) Heatsinks are shown with the wiring depicted in Figure S8 to connect to an LED bar.





### 2.2.5 Light location and characterization setup

To ensure effective and uniform illumination across all wavelengths, the LEDs were strategically positioned to maximize radiative output within the chamber (Figure S3). As we lowered the wavelength, the LED photon flux decreased. To compensate for this reduction, we placed lower wavelength LEDs with relatively lower photon flux directly across from each other. Higher wavelength LEDs with relatively higher photon flux were staggered away from each other (Figure S9). This orientation was chosen to maximize the light coverage of the bag for all wavelengths.

To measure the light output of the LEDs, we used the Ocean Optics Flame-T-UV-VIS spectrophotometer. This spectrophotometer was calibrated for wavelengths of 210-1050 nm. We positioned the cosine receptor in the center of the chamber (Figure S4). We used a tripod setup with the spectrophotometer ziptied onto the tripod and with the cosine receptor clamped down facing towards the measured side (Figure S4). The north side and the south side of the chamber were measured and the values were then summed to obtain the irradiance spectra of the LEDs.

### 2.2.6 Modeling the spectral irradiance

The spectral irradiance at the center of the light box was simulated by modeling the light box as an array of 1440 LEDs positioned at fixed coordinates (Figure 11). The LED radiation patterns provided by the manufacturer were approximated by fitting a sum of cosine-power functions(Moreno and Viveros-Méndez, 2021):

$$I(\theta) = \sum_{i=1}^{3} c_{1,i} \cos\left(|\theta| - c_{2,i}\right)^{c_{3,i}} \tag{1}$$

where $c_{1,i}$, $c_{2,i}$, and $c_{3,i}$ are fitting parameters. We determined that a sum of three terms is sufficient to accurately represent the LED radiation pattern. An example of a measured radiation pattern alongside its fit is shown in Figure S10. The spectral irradiance $E_\lambda$ was then calculated using the following equation (Dragomir et al., 2014) :

$$E_\lambda = \sum_{k=1}^{N} I(\theta_k, \lambda) \frac{\cos\theta_k}{R_k^2}, \tag{2}$$

where N is the total number of LEDs, $\lambda$ denotes the wavelength, $\theta_k$ is the incidence angle for the kth LED, and $R_k$ is the distance from the kth LED to the observation point. The factor $\cos\theta_k$ is computed as:

$$\cos\theta_k = \frac{x - x_k}{\sqrt{(x - x_k)^2 + (y - y_k)^2 + (z - z_k)^2}}, \tag{3}$$

with (x,y,z) representing the coordinates of the observation point (e.g., the center of the light box) and ($x_k$, $y_k$, $z_k$) the coordinates of the $k$th LED. Finally, the spectral profiles of the LEDs were measured directly, enabling us to incorporate the wavelength dependence into the function $I(\theta, \lambda)$ when computing $E_\lambda$ (see Figure 11).





## 2.3 Actinometry

### 2.3.1 Experimental setup

To accurately characterize the LEDs' irradiance within our environmental chamber, we used $NO_2$ chemical actinometry using (Rabani et al., 2021). In the presence of light, NOx and $O_3$, $NO_2$, NOand $O_3$ are in steady-state, being formed and consumed
at the same rate (R1-R3):

$$NO_2 + hv \longrightarrow NO + O \tag{R1}$$

$$O + O_2 + M \longrightarrow O_3 + M \tag{R2}$$

$$NO + O_3 \longrightarrow NO_2 + O_2 \tag{R3}$$

The interconversion of these three gases allows for the measurement of the photolysis rate of $NO_2$ by using Equation 4:

$$J_{NO_2} = k_3 \frac{[O_3][NO]}{[NO_2]} \tag{4}$$

where $k_3 = 1.4 \times 10^{-12} \times e^{(\frac{-10.89}{RT})}$ cm$^3$/molecules s (Atkinson et al., 2004).

For each photochemical experiment, the chamber was first cleaned and then $NO_2$ was directly added to the bag (Praxair 806 ppm $NO_2$ certified standard in $N_2$) until the NOx analyzer measured 40 - 80 ppb. The chamber was then illuminated with the
light source at the maximum intensity (see timeseries in Figure 5).

The actinometry experiments were carried out under three main light conditions: stratospheric (275, 310, 325, 340, 365, 385, 450 - 600 nm), tropospheric (310, 325, 340, 365, 385, 450 - 600 nm), and indoor conditions (365, 385, 450 - 600 nm) (Figure S11 for all experimental timeseries). During these experiments, the lights were switched on until steady-state conditions were reached (Figure 5). When steady state conditions were reached, we used Eq. (4) to calculate the $J_{NOx}$ photolysis rate.

An additional actinometry experiment (Figure S11) was also carried out using fluorescent lights (16 x F32T8-BL) inside a 0.45 m$^3$ perfluoroalkoxy alkane (PFA) bag to compare the actinometry of fluorescent vs LED lighting (Figure S12). The fluorescent lights were mounted to a mobile cart (Figure S12) which provides the flexibility to position the light source in different experimental setups or locations. The same trace gas analyzers were connected to the pillow bag to monitor NOx and $O_3$ mixing ratios during these experiments.






## 3    Results and Discussion

### 3.1    Chamber design and specifications

The ATMOX chamber at UBC is a modular environmental chamber that was designed with features to simulate and study atmospheric photochemical processes across a range of photochemical regimes (Figures 1, 3 and 4). The ATMOX environmental chamber is equipped with a perfluoroalkyl (PFA) 8 $m^3$ cubic bag. The railing system of the chamber bag allows the bag to collapse by pulling the stainless steel rods closer together with a pulley, enabling batch mode and continuous mode experiments (Figure 3). In batch mode, the bag can be filled with a bath gas and the compounds of interest and then contracted without dilution, useful in chemical kinetic experiments. In continuous mode, a steady stream of the bath gas and the compounds of interest are continuously injected and sampled, keeping the volume constant throughout the experiment, useful in SOA generation experiments. Blackout curtains hang from steel rods to cover the chamber and prevent indoor lighting contributing to chemical reactions in the chamber, as well as to protect lab personnel from exposure to the higher energy LEDs (275 nm) (Figures 1b and 3). In addition, the blackout curtains provide unobstructed access to each side of the PFA bag, allowing quick swapping of instruments and tubing lines. The chamber design is further modular; additional components are easily mounted or removed from the frame. The collapsible ATMOX chamber is connected to atmospheric chemistry analytical instruments such as a Vocus proton-transfer-reaction time-of-flight mass spectrometer (PTR-ToF-MS), a scanning mobility particle sizer (SMPS) and trace gas analyzers via FEP tubing (Figure 3). There is also a temperature, relative humidity and pressure sensor connected separately to the chamber with its own 1/2" port (Figure 3). Overall, these integrated features make the ATMOX chamber a powerful tool for conducting controlled environmental chamber experiments.





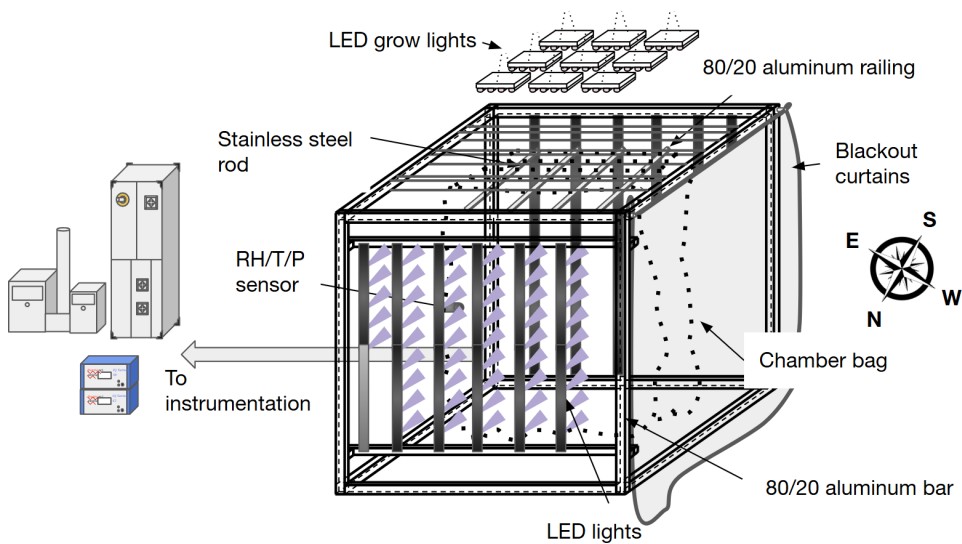

**Figure 3.** The chamber bag is an 8 m$^3$ perfluoroalkyl (PFA) bag with four ports for sampling and injection on two opposite sides of the chamber. The chamber frame is made of 80/20 T slot aluminum bars and supports a pulley system on a stainless steel rod for the bag to collapse in a batch operating mode. One of these ports has a RH/T/P sensor installed (SP-003-1 OMEGA), and the additional ports can be used for the introduction of compounds and for sampling (Vocus PTR-ToF-MS, SMPS and trace gas analyzers are depicted as examples). The 1440 LEDs with wavelengths from 275 to 400+ nm are mounted on bars (12 LEDs per bar) and connected by panels (3 bars per panel). Each bar has a heatsink held by an 80/20 aluminum bar (Figure 2b) and a power control. The outside walls are covered with blackout curtains.

## 3.2   LED setup and irradiance spectra

Individual LEDs have a narrow wavelength range of approximately 20 nm, compared to wider fluorescent light bulbs ranging 100s of nm (Figure 4). Thus, multiple colours of LEDs are required to attempt to reproduce the solar spectrum. The chamber is equipped with a total of 1440 individual LEDs of 6 different wavelengths: 275 nm (72 LEDs), 310 nm (288 LEDs), 325 nm (324 LEDs), 340 nm (324 LEDs), 365 nm (288 LEDs), 385 nm (144 LEDs) (Table 2 and Figure 4). Additionally 1320 commercial LEDs (450 - 630 nm) marketed for indoor plant growth hang from the ceiling of the chamber to supplement the

range of the ATMOX chamber's wavelengths up to 630 nm (Figure 4). These LEDs are more energy efficient and generate less heat compared to fluorescent lighting (Table 3), as well as allow for tunable narrow wavelengths for specific photochemical reactions. The 1440 custom LEDs use a maximum 60-degree lens to provide a wider flux of photons within the bag (Figure S9) (Sharma et al., 2022). The LEDs were mounted on an aluminum panel with passive heatsinks (from Violumas, figure 2b). Passive heatsinks and directed airflow maintain stable temperatures in the chamber with raising experiment temperatures by

$2.40 \times 10^{-3}$ °C per second and 0.8 °C per joule with all the LEDs turned on (equivalent to the stratospheric condition) (see top panel on Figure 5 for time series and temperature date in Table 4).



The irradiance spectrum of the combined LEDs in the ATMOX chamber demonstrates the range of wavelengths accessible with this newer technology (Figure 4). Nonetheless, there are key wavelengths that remain unaccessible with the current LED technology, including 290 nm and 350 nm. The spectrum in Figure 4 represents the maximum irradiance for the LEDs in the ATMOX chamber, but each LED's power can be adjusted to mimic any ratio of intensitity among the different colours of LEDs.

To enable this power adjustment, we designed a potentiometer electrical control panel to adjust the LED power output by changing the current supplied between 20% to 100% (Figure 2a). The choice of the current is thus dependent on the need of the user and the research question. The control panel utilizes a modified MAX472 board (Figure S8). These include cutting the connection between the power supply of the integrated circuits and the current sensing circuit (Figure S7). Additional information on the construction of the control panel are included in the supplementary document and are available upon request. This setup lets the user select any combination of wavelengths, adjust their intensity to modulate reaction rates, and reliably reproduce the same lighting conditions from run to run.

The overall radiative output of the LEDs used in the ATMOX chamber is 364.02 W across all wavelengths, as provided by the engineers at Violumas. Specifically, the radiative output for each wavelength (275, 310, 325, 340, 365, and 385 nm) is 6 W, 23.94 W, 11.88 W, 34.2 W, 144 W, and 144 W, respectively (Table 3). For the quantification of the radiative flux, $J_{NOx}$ chemical actinometry experiments were performed. These combined methods ensured accurate measurement and validation of the LED output for photochemical experiments.




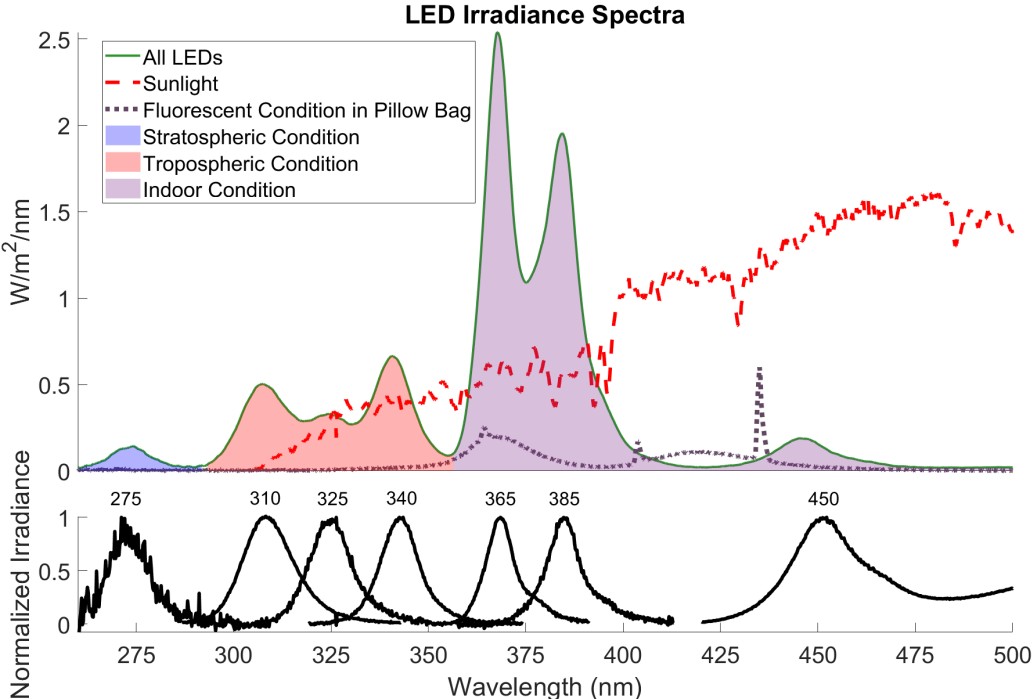

**Figure 4.** The intensity of light is plotted against wavelength for different combination of LEDs. The top panel represents the comparisons between the Violumas LEDs against sunlight and fluorescent lighting. The LED conditions can be split into a stratospheric, tropospheric, and indoor condition. The stratospheric condition includes both tropospheric and indoor light conditions. Similarly, the tropospheric condition includes the indoor light conditions. The bottom half represents the normalized irradiance of each individual LED and their wavelength spread. The sunlight measurement was baseline subtracted to account for the difficulty in taking a completely dark blank outdoors.

## 3.3 Chemical actinometry experiments

### 3.3.1 $J_{NOx}$ photolysis rate experiments

The ability of the LEDs in ATMOX to simulate outdoor sunlight conditions was evaluated by calculating $J_{NOx}$ values from experiments where $NO_2$, $NO$ and $O_3$ were in steady-state concentrations (Figure 5 and Figure S11). During these experiments, $NO_2$ was injected into the 8 $m^3$ chamber and then different combinations of LEDs were turned on until the mixing ratio of $NO_2$, $NO$ and $O_3$ remained stable for approximately 30 mins (Figure 5). We then used Eq. 4 to calculate the $J_{NOx}$ photolysis rates, which are tabulated in Table 3. We also considered three types of combinations of LEDs that we termed indoor, tropospheric and stratospheric conditions as depicted in Figure 4. As expected, the stratospheric conditions provided the highest $J_{NOx}$ photolysis rate, leading to a maximum irradiance of 70.4 $W/m^2$ (Table 3).

Multiple replicate actinometry experiments over 2 years of indoor, tropospheric, and stratospheric LED lighting conditions indicates to us that the photon flux of the LEDs is stable over this time period. Based on the manufacturer, these LEDs have





a lifetime of over 10,000 hours. Each light condition was tested at least three times (Table 3), yielding average $J_{NOx}$ values

of $(4.58 \pm 0.11)$, $(4.84 \pm 0.08)$, and $(4.93 \pm 0.06)$ x $10^{-3}$ $s^{-1}$ for indoor, tropospheric, stratospheric conditions, respectively.

Minimal variation of $J_{NOx}$ values between tropospheric and stratospheric conditions was due to the absorption cross-section

of $NO_2$ which does not extend to 275 nm included in the stratospheric condition (Hawe et al., 2007). These results show the

reliability of photolysis measurements using LED-based lighting conditions.

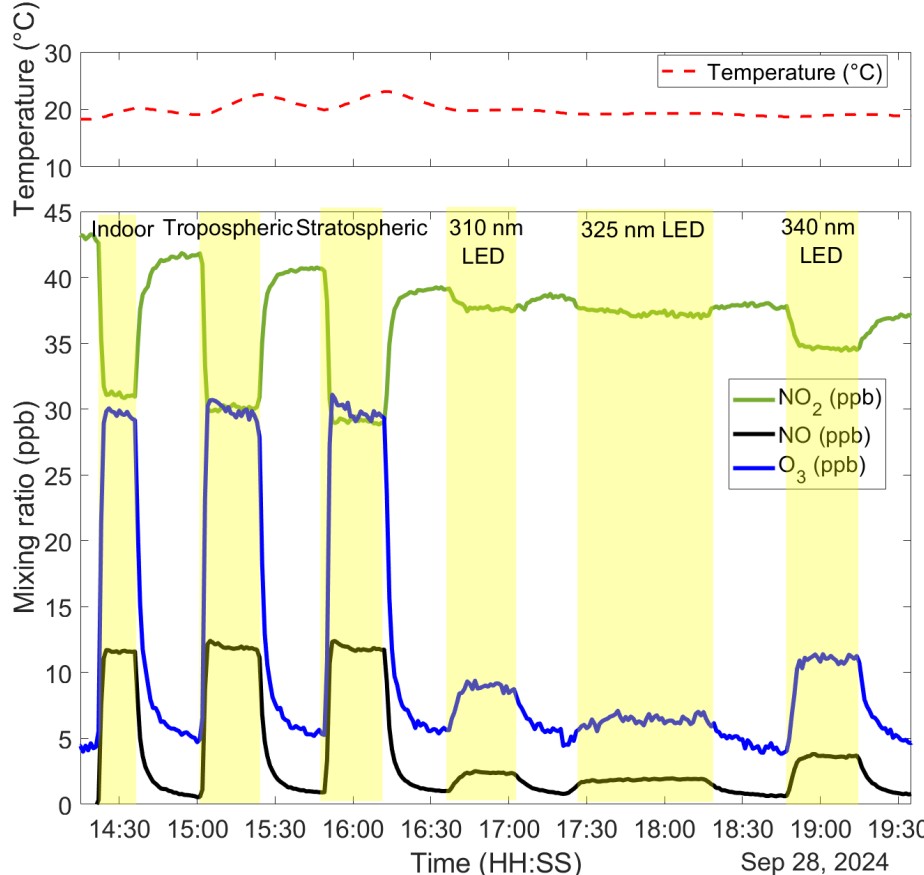

**Figure 5.** Example $J_{NOx}$ experiment of various wavelength conditions. The first photolysis looked at indoor conditions (365, 385, 450 - 600 nm), followed by tropospheric (310, 325, 340, 365, 385, 450 - 600 nm), and stratospheric conditions (275, 310 325, 340, 365, 385, 450 - 600 nm). See Figure S6 for all the experimental timeseries for calculating $J_{NOx}$ values.

### 3.3.2   $J_{NOx}$ value intercomparison

Since $NO_2$ absorbs light over the UV region covered by most of the LEDs we selected as well as most lights used in environmental chambers, the $J_{NOx}$ values serve as a method for comparing different chambers' ability to simulate sunlight-driven chemistry. First, we compared our LED $J_{NOx}$ photolysis rates to the absorption cross-section of $NO_2$ to explain the trend in



increasing $J_{NOx}$ values as a function of wavelength between 310 nm and 385 nm (Figure 6a). Of note, we were unable to mea-
sure the $J_{NOx}$ value for 275 nm or for the commercial plat grow lights, where the overlap between the absorption cross-section
of $NO_2$ and the irradiance is minimal (Figure 6a and Table 3).

We also compared the $J_{NOx}$ of ATMOX LEDs to UV-A fluorescent lighting within our pillow bag setup to further benchmark
the output of the LEDs using the same instruments. The fluorescent light conditions were tested in triplicates, yielding an
average of $(3.28 \pm 0.54) \times 10^{-3}$ $s^{-1}$ (Table 3). These $NO_2$ photolysis rates were comparable to measurements performed
under LED lighting. UV-A fluorescent lights are the standard for simulating atmospheric photochemical processes in other
environmental chambers (Chu et al., 2022) (see Table 1). These findings highlight that LED-based lighting can provide an
alternative to fluorescent systems in atmospheric photochemical simulations without compromise on photon flux output.

To demonstrate the ATMOX chamber's effectiveness compared to ambient settings, we compared $J_{NOx}$ measurements to
real-world outdoor data collected on a rooftop in Vancouver. We sampled during the summer solstice (June 20, 2024 - June
21, 2024), where we obtained a $J_{NOx}$ value of $4.50 \times 10^{-3}$ $s^{-1}$ (Figure 6a). This location was chosen to provide a relevant
comparison to outdoor conditions in Vancouver. These results confirm that ATMOX can photolyze $NO_2$ similarly to real-world
atmospheric conditions.

Furthermore, our measurements show that the $J_{NOx}$ averages in the ATMOX chamber align with reported values from simi-
larly sized chambers using fluorescent lights (Figure 6b) and Table S1). Fluorescent lights from the NC A&T chamber(Smith
et al., 2019) or the BUCT quartz chamber(Ma et al., 2022) reported $J_{NOx}$ of 2.7 to $6.0 \times 10^{-3}$ $s^{-1}$ (additional chamber com-
parisons can be found in Table S1, which also estimates the power consumption). By reporting $J_{NOx}$ values comparable to
those produced by fluorescent lights in other environmental chambers, we further demonstrate the potential of using LEDs for
photolysis in chamber experiments. In addition, the AIDA chamber was equipped with 3013 LEDs ranging from 305 nm to
528 nm and reported a $J_{NOx}$ value of $1.58 \times 10^{-3}$ $s^{-1}$ (Vallon et al., 2022) (Figure 6b and Table S1). These results confirm that
the ATMOX chamber provides $NO_2$ photolysis rates comparable to other sunlit, fluorescent and LED chambers, underscoring
its suitability for atmospheric photochemical studies. This comparison was crucial for validating our experimental setup and
ensuring that our findings could be extrapolated to ambient conditions (Kenagy et al., 2024).



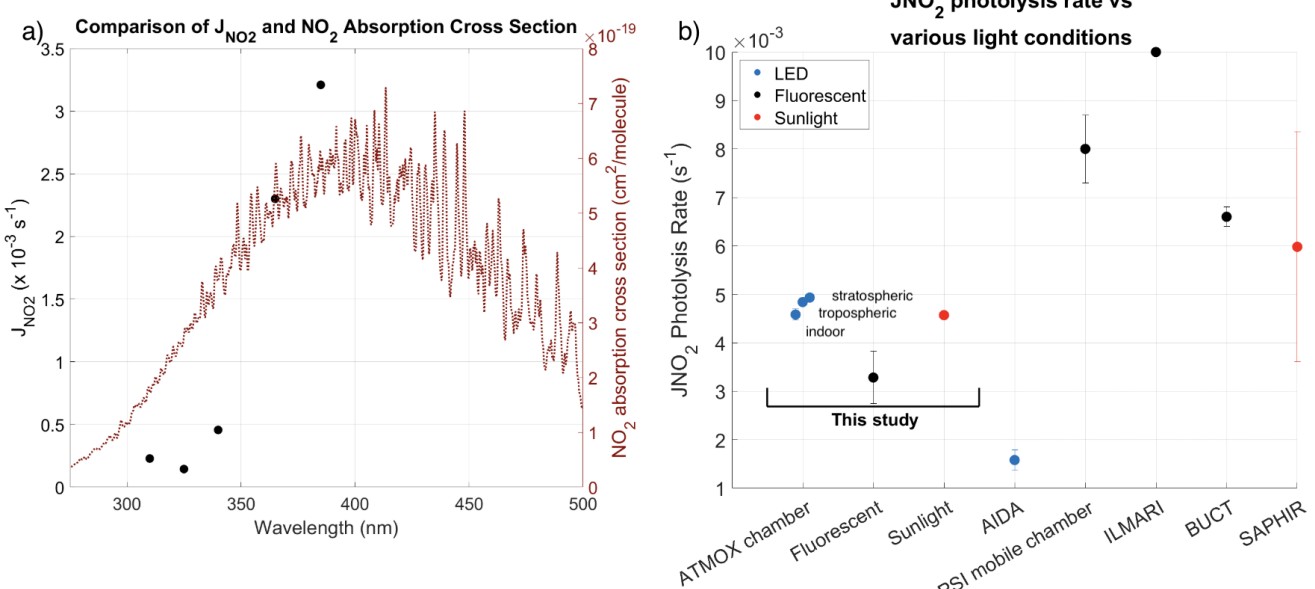

**Figure 6.** (a) This graph compares $J_{NOx}$ values in $10^{-3}$ $s^{-1}$ of the 310 nm, 325 nm, 345 nm, 365 nm and 385 nm as a function of wavelength. 275 nm and the grow lights coudl not be measured quantitatively (N/A). The $J_{NOx}$ absorption cross section data is at 298 K in air from (Burrows et al., 1998). (b) This graph compares the $J_{NOx}$ values measured from different lights in this study: LEDs, fluorescent and sunlight on a rooftop at UBC. We also compare with rates reported for other atmospheric chambers and with ambient sunlight. Blue dots mark LED-illuminated systems, black dots mark fluorescent systems, and red symbols mark natural sunlight. Error bars indicate standard deviation when available. Literature values come from AIDA at KIT, Germany (Vallon et al., 2022), PSI (Platt et al., 2013), ILMARI at UEF, Finland (Leskinen et al., 2015), quartz chamber at BUCT, China (Ma et al., 2022), and SAPHIR in Jülich, Germany (Rohrer et al., 2005).

### 3.3.3 Power consumption

To evaluate the efficiency of the LEDs and compare their power consumption, we measured $J_{NOx}$ alongside the total wattage under each condition (Table 3). The power consumption of ATMOX was comparable to other fluorescent chambers as shown in Table 3. The 385 nm LEDs were the most efficient at photolyzing $NO_2$ per watt followed by the fluorescent light, then the indoor condition and 365 nm LEDs condition. In contrast, the highest $J_{NOx}$ per watt using fluorescent from Table S1 was obtained at $2.6 \times 10^{-6}$ $s^{-1}$ per watt.

### 3.3.4 Photolysis using UV-C wavelength

UV-C radiation, though nearly fully absorbed by the Earth's ozone layer, drives chemical reactions in the stratosphere by creating highly reactive intermediates that influence ozone chemistry and other processes at high altitudes. By incorporating





**Table 3.** Comparison of power consumption, NO$_2$ photolysis rate (J$_{NOx}$), and irradiance across various light conditions. The table details the number of bulbs, total wattage, watts per bulb, J$_{NOx}$ (x $10^{-3}$ s$^{-1}$), J$_{NOx}$ /watt (×$10^{-6}$), and integrated absolute irradiance (W/m$^2$). The indoor conditions use the 365 nm and 385 nm LEDs with the grow lamp LEDs. The tropospheric conditions include the indoor conditions in addition to the 310 nm, 325 and 340 nm LEDs. Finally, the stratospheric conditions include all the LEDs. *Fluorescent light experiments were conducted in a 400 L pillow bag, not in the ATMOX chamber.

| LED | # of LEDs | Total W | W/bulb | J$_{NOx}$ ($\times 10^{-3}$ s$^{-1}$) | J$_{NOx}$ ($\times 10^{-6}$ W$^{-1}$ s$^{-1}$) | Irradiance (W/m$^2$) |
|---|---|---|---|---|---|---|
| 275 | 72 | 317.4 | 4.4 | N/A | N/A | 1.8 |
| 310 | 288 | 1209.6 | 4.2 | $0.228 \pm 0.02$ | 0.186 | 9.2 |
| 325 | 324 | 1242 | 3.8 | $0.144 \pm 0.02$ | 0.104 | 3.9 |
| 340 | 324 | 1242 | 3.8 | $0.456 \pm 0.02$ | 0.349 | 8.8 |
| 365 | 288 | 739.2 | 2.6 | $2.30 \pm 0.01$ | 3.11 | 28.6 |
| 385 | 144 | 362.4 | 2.5 | $3.21 \pm 0.02$ | 8.86 | 24.4 |
| grow lights | 1320 | 540 | 2.4 | N/A | N/A | 10.1 |
| Indoor | 1752 | 1641.6. | 0.6 | $4.58 \pm 0.11$ | 2.78 | 62.7 |
| Tropospheric | 2688 | 5335.2 | 1.8 | $4.84 \pm 0.08$ | 0.907 | 83.8 |
| Stratospheric | 2760 | 5652.6 | 1.9 | $4.93 \pm 0.06$ | 0.872 | 85.2 |
| Fluorescent* | 16 | 512 | 32 | $3.28 \pm 0.54$ | 6.40 | 10.6 |

275 nm LEDs into the ATMOX chamber, we recreate this energetic UV environment under controlled laboratory conditions,
enabling the study of stratospherically relevant photochemical pathways.

Since the absorption cross-section of NO$_2$ does not extend into the UV-C region, a JNO2 value could not be measured for the 275 nm LEDs (Figure 6a). Hence, we used 2-nitrobenzaldehyde as an actinometer to showcase the effective irradiance of the 275 nm LEDs and simulated stratospheric photochemical conditions (Bouya et al., 2017; Xiang et al., 2009). Furthermore 2-nitrobenzaldehyde is atmospherically relevant in the context of biomass burning organic aerosol penetrating into the
365 stratosphere (Kahnt et al., 2013). Despite comprising only 72 individual LEDs, the 275 nm condition delivers sufficient UV-C flux to effectively drive the photolysis of compounds relevant to stratospheric chemistry. These LEDs effectively photolyze 2-nitrobenzaldehyde (Bouya et al., 2017; Xiang et al., 2009), with a k$_{observed}$ of $3.47 \times 10^{-4}$ s$^{-1}$ (Figure 7). This result confirms that even a compact 275 nm LED system can reliably simulate stratospheric photolysis for controlled photochemical experiments.

Overall, these findings demonstrated that 275 nm LEDs can effectively photolyze compounds of stratospheric interest, highlighting the LED-based approach for replicating energetic processes typically confined to the stratosphere. By combining high-intensity UV-C irradiation with a bottom-up method that targets individual chemicals in the ATMOX chamber, we can refine our understanding of how these species transform, ultimately deepening our insights into key stratospheric chemical mechanisms.




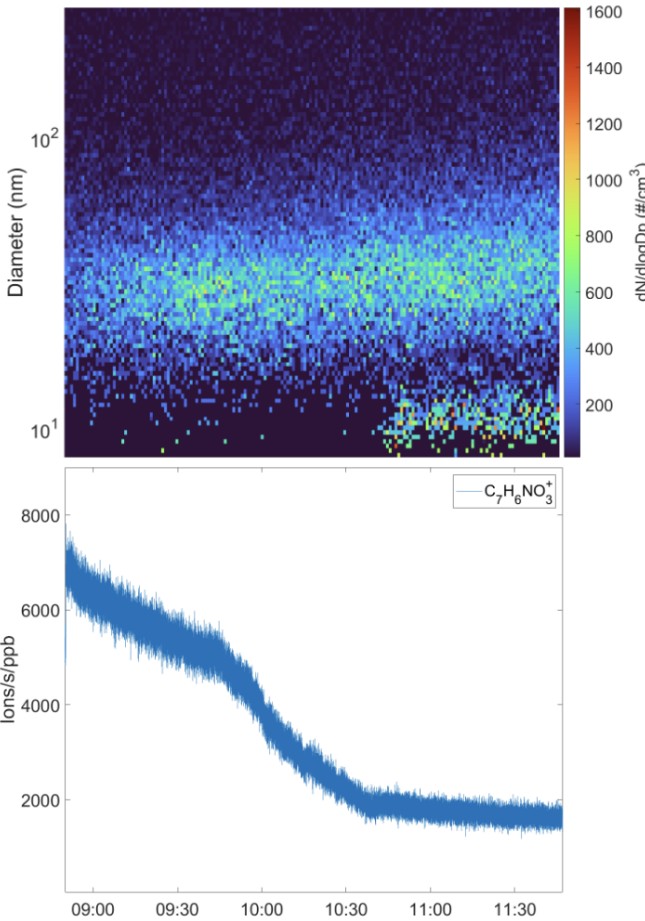

**Figure 7.** Measurements during the photolysis of 2-nitrobenzaldehyde. The top panel shows how the number of particles at different diameters changed over time; brighter colours mean more particles at that size. The bottom panel plots the mass signal for the parent ion $C_7H_6NO_3^+$; the shaded band marks the period when the 275 nm LEDs were switched on. The signal drops while new particles appear, indicating that the compound is being photolysed and some of the products are forming aerosols.

## 3.4 Chamber operational characteristics

### 3.4.1 Temperature differences between fluorescent and LEDs

The temperature measured by the temperature sensor connected to the PFA bag to compare the heating rate of both the LEDs and UV-A fluorescent lighting (Table 4). The temperature was monitored during irradiation experiments (see Figure S11 for the time series and Table S2 for the values). Then a temperature ramp per second was calculated, indicating that the LEDs increased the temperature inside the temperature by a slower rate, unless all 2760 LEDs were turned on (Table 4). Nevertheless, if we compare the temperature increase per joule, then the fluorescent lights are calculated to produce an order of magnitude higher temperature increase rates per joule than the LEDs (Table 4). These rates highlight the differences in the amount of power



each light source used and their corresponding temperature increase within the chamber. In all, the LEDs produce less heat compared to UV-A fluorescent lighting on a per joule comparison (Table 4).

**Table 4.** Summary of temperature changes under a range of light conditions. Each row shows the total temperature change ($\Delta T$) between the initial temperature and the peak temperature during illumination, the wattage applied, duration in seconds of the temperature increase, the calculated temperature increase (ramp) rate per second, and the temperature increase (ramp) rate per joule.

| Light condition | $\Delta T$ | Input wattage (W) | Condition length (s) | Temp ramp per sec | Temp ramp per joule |
|---|---|---|---|---|---|
| 310 | 0.1 | 1209.6 | 1380 | $7.2 \times 10^{-5}$ | 0.1 |
| 325 | 0.1 | 1242 | 2940 | $3.4 \times 10^{-5}$ | 0.2 |
| 340 | 0.5 | 1242 | 1560 | $3.2 \times 10^{-4}$ | 0.6 |
| 365 | 2.1 | 739.2 | 2700 | $7.8 \times 10^{-4}$ | 7.7 |
| 385 | 0.9 | 362.4 | 2580 | $3.5 \times 10^{-4}$ | 6.4 |
| Indoor | 1.8 | 1101.6 | 1020 | $1.8 \times 10^{-3}$ | 1.7 |
| Tropospheric | 3.5 | 4795.2 | 1500 | $2.3 \times 10^{-3}$ | 1.1 |
| Stratospheric | 3.2 | 5112.6 | 1320 | $2.4 \times 10^{-3}$ | 0.8 |
| Fluorescent | 3.5 | 512 | 3360 | $1.0 \times 10^{-3}$ | 22.9 |

### 3.4.2 Wall loss in the chamber

The wall loss of gases and particles inside an environmental chamber is an important drawback and needed to be accurately quantified for kinetic experiments, for example (Wang et al., 2018; Krechmer et al., 2020). In our chamber, we measured the rate of the wall loss of $NO_2$ to be $3.9 \times 10^{-6}$ s$^{-1}$ (Figure S13a). The greater the amount of surface available, the more likely the compound in the chamber will be lost to the wall, and diffuse through the chamber walls. This size could become problematic with smaller chambers as some chemical systems required long periods of time to react and stabilize. For example, in our group, the ozonolysis of cashmeran required the 8 m$^3$ ATMOX chamber due to the long lifetime of cashmeran (Akande and Borduas-Dedekind, 2025). Indeed, during preliminary experiments in a 450 L pillow bag, the wall loss of cashmeran was faster than its reaction time with ozone. On the other hand, also in our group, the ozonolysis of methylated selenium compounds involved much faster reactions, making a smaller 450 L PFA bag sufficient for these kinetic experiments (Heine and Borduas-Dedekind, 2023).

### 3.4.3 Cleaning of the chamber

Additionally, cleaning the chamber can be typically done by purging the PFA bag overnight with a source of clean air or $N_2$ (Figure S13b). However, following the actinometry experiments with $NO_2$, it was necessary to clean the chamber more effectively by using a pressurized wash system (ECHO MS-21H) filled with MilliQ water (18.2 ohm). The resulting $NO_2$





background mixing ratios were then below the detection limit of the gas analyser (Figure S13c).

## 3.5 LED Light Modeling for future users

### 3.5.1 Irradiance output within the chamber

To better characterize the ATMOX chamber, we developed a modular model to enable the optimization of LED positioning and provide insights into the behavior of the light output within the chamber for future users (Figure 8). The model predicts a distribution of irradiance between 1000 and 5000 $\mu$W/cm$^2$ with all the LEDs turned on to maximum power. The center of the chamber receives the most photons, decreasing in a concentric circle as a function of distance (Figure 8a,b). It is notable that the LEDs are powerful enough to emit irradiation beyond a 1 m distance, while the normalized intensity decreased exponentially

by approximately 90% over 1 m (Figure S6). Furthermore, we confirmed that the PFA bag does not absorb light by repeating the irradiance measurements as a function of distance in the absence and presence of the PFA bag (Figure S6). A decrease of only 6% was observed, indicating that the light transmission through the PFA bag is efficient (Figure S6). The spectral irradiance can also be calculated from the predictive model (Figure 8c) and matches within a factor of 2 to the sum of the measured irradiance from the four corners of the ATMOX chamber (Figure 9). By simulating light distribution, the model

helps improve the uniformity and efficiency of photochemical experiments.

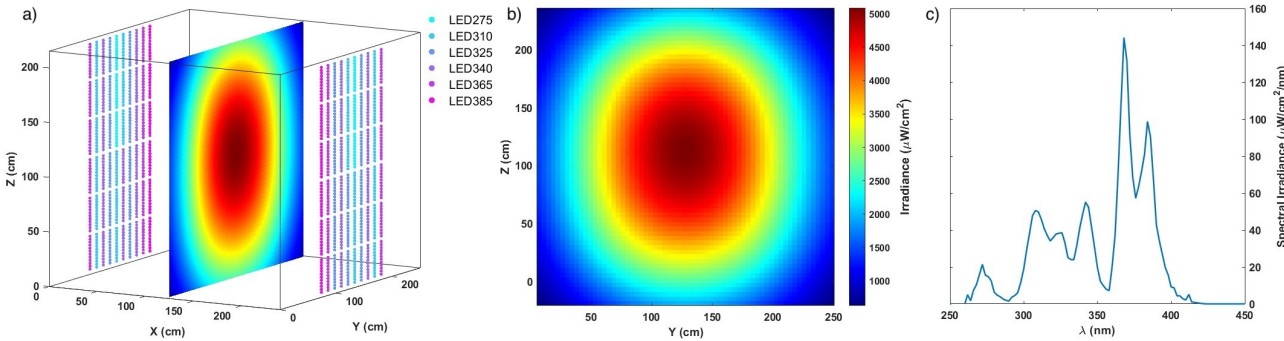

**Figure 8.** (a) Simulation of the LED light box shows 1440 LEDs arranged in 120 arrays, with each array consisting of 12 LEDs. The colour-coded legend represents the wavelength assigned to each LED array. The heat map displays the irradiance on a plane located midway between the LED panels, while figure (b) presents the spectral irradiance of the LEDs at the central position. (c) The predicted irradiance is shown as a function of wavelength.





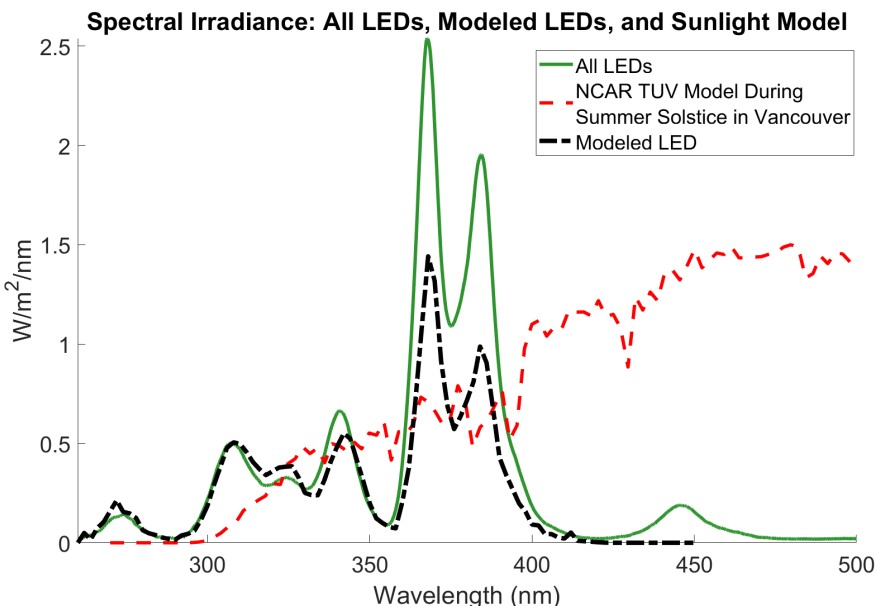

**Figure 9.** The spectral irradiance of the LED array was measured inside the chamber by summing the north and south panel irradiance (solid green), and is compared to the spectrum predicted by the ATMOX-based computational model (black dash-dot). For context, the NCAR-TUV clear-sky solar spectrum for the summer solstice in Vancouver is also shown (red dashed). The modeled LED trace reproduces the magnitude and shape of the dominant UV peaks between $\approx$ 300–400 nm. All curves are presented as absolute spectral irradiance (W/m$^2$/nm$^1$) over the 260–500 nm wavelength range.

### 3.5.2 Estimating the number of LEDs needed

Based on our experimental measurements of actinometry for the photolysis of NO$_2$ into NO and O$_3$, we developed a model that translates these wavelength-specific rates into a predictive framework. This model allows us to simulate the photolysis environment of our current chamber and assess how LED count and arrangement affect overall irradiance (Figure 8). By

exploring these relationships, we are able to establish a guideline for scaling the LED configuration in future setups, ensuring that they replicate the desired photolysis conditions effectively. This model can also be used and consulted upon request for community members to estimate the number of LEDs necessary in their chamber of any size.

## 4 Conclusion and outlook

Environmental chambers are continually evolving, and advancements in LED technology offer promising potential for even

more efficient simulation of atmospheric chemistry. As LED output at UV wavelengths improves, we can better replicate not only tropospheric but also stratospheric conditions, providing more accurate insights into photochemical processes. Our findings, with comparable J$_{NO_x}$ values across different light conditions and setups, demonstrate that our chamber is well-suited for such studies. By refining our approach and leveraging these technological advancements, we can deepen our understanding



of the complex interactions within our atmosphere, ultimately enhancing our ability to predict and mitigate the impacts of
atmospheric pollutants on air quality, climate, and global atmospheric chemistry.

*Data availability.*  All information not included in the SI is available upon requesting.

*Author contributions.*  All authors contributed to the design of the chamber. RJML and NBD led the design, purchasing, construction and
validation of the LEDs and the chamber. RJML ran the actinometry experiments. AAA was the first user of the chamber and developed
protocols with RJML. SK developed the program for modeling and simulating the LED lights. PAH ran the pillow bag experiments with
fluorescent lights. PP built the structure of the chamber. DT and WR provided support in testing, mounting, and maintaining the LED setup.
MR assisted with organizing and designing the LED setup. NBD obtained the funding and supervised the project. The manuscript was written
by RJML and NBD with contributions from all authors.

*Competing interests.*  The contact author has declared that neither they nor their co-authors have any competing interests.

*Acknowledgements.*  We are grateful to NSERC Research Tools and Infrastructure for funding. We acknowledge Violumas for providing
technical insights on the UV LEDs, and Ingeniven for providing comments on the design of the PFA bag. We also acknowledgment initial
design contributions from Des Lovrity and Tony Mittertreiner. We also thank Dylan O'Leary and Christopher Manke for recent help with the
assembly of the electronics.



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
