# Peer review of "The UBC ATMOX chamber: An 8 m3 LED-powered modular environmental chamber for indoor and outdoor atmospheric chemistry"

_EGUsphere, 2025_

## Referee Comment (RC1)

**General comments:**

The preprint by Lee and co-workers presents the design and implementation of the UBC ATMOX chamber - an innovative, modular 8 m³ environmental chamber. A central aspect of the manuscript details the design of a light system for the chamber, based on LEDs, enabling wavelength-specific research on photochemical processes of aerosol particles and gases. For instance, the system enables experiments under light conditiosn that are simulating stratospheric, tropospheric, and indoor light conditions, as detailed in the presented work.

While many different atmospheric simulation chambers exsist around the world, I find the design and use of an LEDs light system for an atmospheric simulation chamber as presented here a very interesting idea. I believe this concept has strong potential to influence future chamber designs and thus clearly aligns with the scope of *Atmospheric Measurement Techniques*. That said, the manuscript in its current form contains numerous (minor) inconsistencies that detract from its clarity and impact. I encourage the authors to carefully address the points listed below, along with the more detailed comments provided. Given the number and nature of the technical suggestions, I recommend a thorough revision followed by re-assessment for publication.

**Specific comments:**

Title: Superscript "3" in m$^3$

Abstract: The abstract would benefit from being shortened a bit and focus on the bigger picture of the manuscript. E.g.. consider removing details such as the overall costs of the LEDs or the specifications of the T-slots used to collapse the chamber. For the latter, the description used on e.g., L121 "It is also equipped..." seems more appropriate for an abstract, and also in view of the fact that the squeezability of the chamber is not discussed much in the main part of the manuscript.

L11: change "LED grow plant lights" to "LED plant grow lights" for consistency with the remainder in the mansucript.

L44: Formatting of reference, i.e., write as: "Nguyen et a. (2023) have …"

L46: Please introduce abbreviations of GEOS-CHEM and F0AM or generalize to "global climate and box-model simulations" or similar.

L47: change to : "peroxy radical (RO$_2$)"

L55: Consider other chambers with temperature control, e.g., (Kristensen et al., 2020; Zong et al., 2023).

L70: This reference seems inappropriate, at least there are many other studies, where the material collection from atmospheric chambers is the focus of the paper. You might want to have a look at e.g.: https://link.springer.com/book/10.1007/978-3-031-22277-1, which might in general be of interest for your paper.

L79: RO2 has been introduced above (L47), but the other have not. Consider to properly introduce OH and HO2 here, or when first mentioning these.

L80: Please define "FEP". For the chambers, none of the acronyms seem to be defined in the text. While this is probably okay to improve readability, consider adding the definition of each acronym either to your Table 1 or in a list of acronyms at the end of the paper.

L82: consider saying "in China", to be consistent with wording on L80-81, where you state countries instead of cities.

L83: replace "simulations" by "studies"

L86: delete "proton"

L87: In Table 1 you write as "CaltTech", please use one of these consistently throughout.

L89: change to "applied to study e.g.. cellular..." to account for the many other types of studies that have been conducted at this facility.

L91: Please define what you mean with "UV blacklights".

L92: Subscirpt "x" in "$NO_x$" here and elsewhere in manuscript.

L93: Change to "These examples…"

L94: delete "atmospheric" as you also cover examples of chambers used to study indoor chemistry.

L97: Does the UPatras chamber have a name? For all other examples in this sentence you use the chamber name, rather than the institute where it is located.

Table 1: Caption:

- Please add "some environmental chambers", to clarify that this list is not complete, but rather presents a list of chambers that have inspired the current chamber design.
- Consider changing the column title from "Characteristics" to "Key characteristics" or similar to again make clear that the listed features are examples. The lists of characteristics seems incomplete and or the choice of characteristics listed here unclear and inconsistent. For example, some of the other chambers also have the ability to control/vary T.
- UBC, Canada chamber: How is T-controlled in your chamber? Looking at Fig. 1b, it seems like the chamber T is determined via the temperature setting in the room. In addition, without quantification of e.g. the T-range and the accuracy of the T-control the wording of "precise T-control" seems inappropriate here and elsewhere in the table. Please adjust. Related, looking at some of your $\Delta T$ values reported in Table 4, I would not call it "precise T control."

L130: While I appreciate the technical detail, I feel that the readability of text would benefit from removing some of the detail. E.g., the type of screws can probably be omitted. In addition, introduction of "BHSCS" seems unnecessary, as it is not further used in the text below. Please consider improving.

L140: rewrite to: "from light emitted from the 275 nm LEDs."

Figure 1:

- Panel d: The grow lights are hardly visible on this photograph. Consider exchanging this panel with a photo that more clearly shows the different light types, as this is key to your presented manuscript.

L143: change to: "…a scanning mobility particle sizer (…) was used to measure particles from 2 - 1000 nm."

L149: "Additional instrumentation can selectively be coupled to the chamber to monitor other parameters, including a relative ..." Please also give details on company and model of the RHTP sensor used. Clarify why it was wrapped in PTFE or omit this part of the sentence.

L154: change to: "by aerosols and gases" and "all directions,"

L159: add comma after "washer"; please check units, should be "18.2 M $\Omega$ cm"

L169: Entire paragraph: This text can be improved. Consider removing the costs from the text and just keep it in Table 2. Just having a total costys as here in the text without further information on the type/model, and number of LEDs is not very useful.

Table 2:

- Caption: Delete "All totals…"
- Rows: Here it could be helpful to include e.g., the part number or similar for each LED, to enable other researcher to possible reproduce your system. This could also help to link each LED listed here to your wording used below, e.g.., on L178 "The UN-C LED..."
- Write as either "LED" or "LEDs", right now you have a mix.

L178: "The UV-C LED (VC1…)" move the part number/specification to the table and omit in the text to improve readability.

L185: Please introduce acroynm as "electrostatic discharge (ESD)", and remove from L188.

L217: Do you mean chamber bag with "light box"? Please specify, as you use is throughout. I can also not find a Figure 11, please correct (also L230).

L219: Add space fater "functions"

L221: Please define all parameters of equation, i.e., also "ө" and "I". Is I the relative intensity that you show in Fig. S10?

L233: This sentene appears incomplete, please check.

L234: Subscript in "NOx"

L242: Please write units as "$cm^3$ $molecules^{-1}$ $s^{-1}$" for consistency; see also AMT style guide.

L246-248: Can you please provide some reasoning or references why the combination of the listed LEDs represent "stratospheric", "tropospheric" and "indoor light" conditions? It could also help to follow parts of your discussion to add these definitions of "stratospheric", "tropospheric" and "indoor" to your Table 2.

L262: do you mean "filled with a batch of gas"? Please check use of "bath gas" here and on L263.

L263: add "useful e.g., in chemical"

L264: add "useful e.g., in SOA"

Sect. 31.: Several of the aspects described here are already part of Sect. 2.1. E.g., the material type of the bag is already mentioned on L132. The fact that the black out curtains inhibit unwated photochemistry from room light and are protecting people working in the lab, is already mentioned on L139. The instrumentation (e.g. VOCUS) attached to the chamber is already mentioned on L144. It would be good to remove some of this redundancy to improve readability.

L268: What "additional components" are you thinking of? It would be good to name some examples here.

L272: "Overall, these integrated features…". Are these really "intergated" or are these instruments that can selectively be attached to the chamber, depending on the instruments needs? I like your description in the caption of Fig. 3 better. Consider to revise.

Fig. 3: Caption: I feel that some of the technical specifications could be removed from the caption, e.g., the type of RHTP sensor ("SP-003-1 OMEGA") or the type of aluminum bar ("80/20") seems unncessary and repetitiev here. Please also note different typing of "RH/T/P" here and "RHTP" on e.g. L150. I also recomment to state the LED wavelengths as "275-385 nm" and then state the wavelength of the growth lights separately; the use of "400+ nm" seems odd. You could also simply refer to your Table 2 that contains all the specifications, to make the caption of your Fig. 3 more succinct.

L276: change to "(Figure 4b)"

Sect. 3.2: This section should be shortened, as a lot of the information has already been provided. E.g., your Table 2 already specifies the number of LEDs used for each wavelength. The "60° lens" is already described

on L176. It would be helpful to focus the attention in this "Results and Discussion" section more on the actual data that you show e.g., in your Figs. 4 and 5, and discussion thereof..

- L279: delete "commercial"
- L282: change to "60°"
- L283: capitalize "Figure 2b"

Fig. 4, 5: Please introduce panel labels, e.g., (a), (b) and refer to these in the caption and text, rather than using "top" and "bottom", to be consistent with the journal guidelines. As you have done for some of your other figures, e.g., your Fig. 6.

Fig. 4: What does "Pillow Bag" in your legend correspond to?

L284: I would not describe this as "stable temperature". Your timline in Fig. 5a shows distinct fluctuations, e.g., around 16:00, labelled "Stratospheric" (same in top panels of your Fig. S11). Looking at your values listed in Table 4, it seems like over the 22 min experiment of stratospheric conditions, the chamber temperature increased by 3.2 K, which is similar to the temperature increase observed with fluorescence light (over a longer time period). I might be miss-interpreting something here, but your statement further down (L380) suggests that the temperature inside the chamber was not stable with all LEDs on. Please explain better in the text what you consider "stable" . It could also help the structure of the manuscript to move the discussion of heatsinks and stability of chamber temperature (L283-286) to Sect. 3.4.1.

L290: change to "intensity ratio" or "ratio of intensities"

L291-293: I really like your idea to include a potentiometer to control the power of each LEDs. I am, however, unclear what the LED power settings are in your Figs., e.g., 4 and 5. Were the LEDs always fully powered unless specified? Please add this information to your text and/or figure captions.

L296: change to "… while reliably reproducing the same same light condition between independent experiemnts, if desired."

L299: rephrase to "…output is 6 W, 23.94 W, …, at wavelnegth of 275, 310… and 385 nm, respectively".

L300-302: "For the quantification…". Consider moving this sentence to the beginning of Sect. 3.3.1 to improve structure.

L309: Eq. (4) depends sensitively on the temperature inside the chamber. How were temperature fluctuations (e.g., top panel in Figs. 5 and S11) handled in calculating $J_{Nox}$?

L310: These three categories "stratospheric, …" should be introduced more prominentely earlier in the manuscirpt. Could you add the wavelength of the LEDs that were turned on for each of these categories in

your Fig. 4? That could help to clarify that your stratopsheric conditions were not limited to below 300 nm, as the coloring in your Fig. 4 suggests, but cover 275-600 nm, as you write on L246.

L311: Add comma before "as depicted"

L315: What is the typical lifetime of UV-A fluorescent lights that are traditionally used in chamber setups? Adding such a value here would be handy for comparison.

L318: Add comma before "which"

L321: Add comma after "we selected"

L323: Add comma after "of NO2"

L325: "plant"

L327: The "pillow bag setup" that is mentioned here and also already appears in Fig. 4 is confusing. Is this pillow bag connected to the ATMOX chamber? This pillow bag setup and how it compares or is different to the ATMOX chamber should be clarified. This information could be added to the "Methods" section (maybe in its own subsection), after it is first mentioned on L253. Further below on L392 you state that the pillow bag has a volume fo 450 L. This is much smaller than the ATMOX chamber, and the irradiation conditons will be different. I am not fully clear how you can then directly compare the NOx photolysis rates between these two setups? Please expand on this in the text.

L335: Shouyld this not be "(Figure 6b)"?

L339: Change to "(Figure 6b and Table S1)" add space after "chamber" here and on L340

Figure 6:

- Please change the x-axis label of the third item from the left from "Sunlight" to "UBC rooftop outdoor measurement" or appropriate, as the red color of the dot already indicates "Sunlight".
- Consider removing the labels on top of the panels, as this is what you describe in the figure caption. Also writing of "$J_{NO2}$" vs. "$JNO_2$" is inconsistent, and the general use of "$J_{NOx}$" vs. "$J_{NO2}$" is inconsitent throughout the text; please fix.
- Caption:
  - fix typo "could not be"
  - change to "… air from Burrows et al. (1998)"
  - Please clarify in the caption if the error bars denote one standard deviation, or other.

Table 3:

- Please at units of "nm" to the column "LED"

- Please check volume of pillow bag, which is stated as 400 L here, but as 450 L on L392.
- Units of "Irradiance" should be changed to "W m$^{-2}$" for consistency.
- As mentioned above, it could be helpful to add an overview of the different LEDs used for "stratospheric", "tropoospheric" vs. "indoor" conditions furtehr up in the mansucript. Then you could just refer to this specification, rather than repeating this information here and elsewhere (e.g., caption of Fig. 5). For instance, you could add 3 columns for "stratospheric", "tropoospheric" vs. "indoor" to your Table 2, and then check all the LEDs that were used for a given conditions.
- Caption: add space, i.e., write "$\times$ 10$^{-6}$" and use the same symbol for "$\times$"
- Caption: Do the uncertainties also correspond to one standard deviation? Please clarify in the caption.

L357: Please add appropriate reference to this statement.

L361: Please make formatting of "JNO2" consistent throughout text and use subscript, i.e., "$J_{NO2}$"

L362: Please change to "Hence, we used photolysis of 2-nitrobenzaldehyde to explore the irradiance of the 275 nm LEDs to simulate photochemical conditions in the stratosphere (refereces)."

L366: "These LEDs effectively… with a $k_{observed}$ of…" The disucssion of Fig. 7 is insufficient. Please expand the explanation of remove the figure from the text. Fig. 7a is not discussed at all. Furthermore, it is unclear to me how the stated $k_{observed}$ can be derived from Fig. 7b. The text discusses photolysis of 2-nitrobenzaldehyde, and Fig. 7b shows the temporal evolution of the parant ion C7H6NO3+, but these should be connected/explained better in the text.

Figure 7:

- Please add panel labels (a) and (b).
- Caption: I do not see "shaded bands" that mark LEDs to be turned on.
- Caption: Specify, as "The gas phase signal of … decreases while new aerosol particles are formed, indicating…"

L379: Here and elsewhere, consider using "temperature change per second" rather than "temperature ramp per second"

L377: "to compare the heating taye of both the LEDs and UV-A fluorescenct lighing…". It is unclear to me if both types of light sources were measured at the ATMOX chamber, or if the fluorescence lightning was used together with a pillow bag setup, as done above (e.g., your Table 3 and Fig. 4). Please clarify in the text.

L380: "… unless all 2760 LEDs were turned on (Table 4)." Where do I see this value in your Table 4? , This corresponds to the "stratospheric conditions", correct?

Sect. 3.4.1: The discussion of the temperature increase inside the ATMOX chamber, seems to be based on the measurement of a single T-sensor connected to the side of the PFA bag (L378). What about temperature mixing/homogeneity inside the chamber volume? Is this single point measurement representative? Please add discussion thereof.

Table 4:

- Units are missing for $\Delta T$.
- Caption: change "total temperature change" to "absolute temperature change"

L387: change to: " for kinetic experiments (e.g., Wang …)."

L388: "The greater the amount of surface available…" Is this not an effect of surface-to-volume ratio of laboratory chambers? Please specify your discussion.

L389: delete "and diffuse through the chamber walls" or provide further explanation.

L389: "This size…" on L389 should probably be changed to: "Small chamber vloume, and associated larger surface-to-volume ratios, could limit smaller chambers, as some …"

Sect. 3.4.2. This section is focused around one value, the wall loss of NO2. However is unclear to me if the wall loss of gas species and aerosol particles directly comparable? About half of this section describes pillow bag experiments that are part of other publications of the group. I encourage the authors to provide some more details and discussion of the wall loss characterization of the ATMOX chamber, or move this section to the SI along with Fig. S13a.

Sect. 3.4.3 only extends over 4 rows. I encourage the authors to merge the content of Sect. 3.4.3 into Sect. 2.1.3, stating that the chamber can be effectively cleaned either by a purge flow or by washing. This would further help to focus the "Results" section of this paper more on the lighting system of the chamber and characterization thereof, which is the central aspect of the work presented.

L400: Units of MilliQ water resistance are wrong.

L407: Write units as "$\mu W\ cm^{-2}$"

L407: "all LEDs". Does this model really take into account ALL LEDs, including the grow lights? Looking at your Fig. 3 and Fig. 8a, it appears to me that for the model presented here only captures LEDs mounted along the side walls of ATMOX, i.e., LEDs with $\lambda \leq 385$ nm. Please clarify. If the grow LEDs were to be included, I would expect the irradiation distribution shown in Fig. 8b to not be perfectly center-symmetric. Note that this also affects the legend and description of Fig. 9.

L414: Which "four corners"? If I count the corner of your cubic chamber, there should be 8.

L413: Looking at your Fig. 8c, what x,y,z position in the chamber (your Fig. 8a) does this correspond to?

Fig. 9: I am not clear what the purpose of the red, dashed line is? It is not discussed in the text, which is focussed on a comparison of the measured vs. modelled irradiance of the ATMOX chamber.

Sect. 3.5.2: Based on the section title, I would have expeced some senstivity studies on how different numbers and placements of LEDs affect the irradiance in the ATMOX chamber, or an example that applies this model to having more/less LEDs, LEDs of different wavelengths, or chamber of different type (volume). Having this model will certainly be very beneficial for the community. However, I feel that the discussion thereof in its current form remains somewhat superficial. Thus, I would encourage to merge the relevant aspects of this very short Sect. 3.5.2 with Sect. 3.5.1, or alternatively provide some more in-depth discussion of your model.

L424: Rephrase to: "… offer great potential for improved simulation of photochemical processes relevant for Earth's atmosphere."

Section 4:This conclusion and outlook section remains rather vague. Upon revising their manuscript, I would encourage the authors to rework this section a bit. E.g., you could include the potential use of your irradiaation model presented in Sect. 3.5 and expand on your plans to use it to simulate different tropospheric environments ("locations") in your chamber, or further plans to add LEDs of other wavelengths, if applicable. In addition, the authors could maybe provide some thoughts on how the LEDs can be used to study wavelength-dependent photochemical reactions, something that I regard as very powerful and a unique feature of your setup.

Kristensen, K., Jensen, L. N., Quéléver, L. L. J., Christiansen, S., Rosati, B., Elm, J., Teiwes, R., Pedersen, H. B., Glasius, M., Ehn, M., and Bilde, M.: The Aarhus Chamber Campaign on Highly Oxygenated Organic Molecules and Aerosols (ACCHA): particle formation, organic acids, and dimer esters from $\alpha$-pinene ozonolysis at different temperatures, Atmospheric Chemistry and Physics, 20, 12549–12567, https://doi.org/10.5194/acp-20-12549-2020, 2020.

Zong, T., Wu, Z., Wang, J., Bi, K., Fang, W., Yang, Y., Yu, X., Bao, Z., Meng, X., Zhang, Y., Guo, S., Chen, Y., Liu, C., Zhang, Y., Li, S.-M., and Hu, M.: A new smog chamber system for atmospheric multiphase chemistry study: design and characterization, Atmospheric Measurement Techniques, 16, 3679–3692, https://doi.org/10.5194/amt-16-3679-2023, 2023.

---

## Author Comment (AC2)

Notes:
Referee  comments are in black
Author responses are in blue
*Modified text in the revised manuscript is in blue italics*

Referee #2:

General comments

Lee et al. describe the setup of a new environmental simulation chamber with collapsible Teflon walls and flexible illumination by different types of modern LED. The chamber design is described in detail with emphasis on the LED choice, setup and operation. The overall concept is sound and the advantages of using flexible LED illumination are convincing. Also, the comparison of basic properties of this new chamber with several existing facilities worldwide is useful.

We thank the reviewer for their appreciation of our LED design and our intercomparison with other chambers.

However, the paper is not well structured. Sections 2 and 3 contain partly redundant information and recurrent topics. Many technical details could be shifted to the supplement to improve the relevance for the scientific audience.

We agree with the reviewer and we have made major revisions to sections 2 (methods) and 3 (results) to avoid redundancy, including moving more technical details to the SI (including the LED wiring, current and voltage details). We have now followed the structure of the Vallon et al. AMT 2022 LED AIDA chamber paper where we have many subsections, but none called explicitly methods or results to help with the flow.

My main criticism refers to the analysis of the obtained data which is insufficient. The authors determined $jNO_2$ in actinometric experiments but the relationship between the measured LED irradiance spectra and the actinometric $jNO_2$ is merely discussed on a qualitative level. What these results mean for the photolysis of other species that will sooner or later be used in this chamber remains unclear. The authors do not seem to be aware of the difference between spectral actinic flux and spectral irradiance or do not discuss it adequately. These quantities are not exchangeable as implied by the $jNO_2$ determined from the outdoor irradiance measurements. Moreover, there is an inconsistency in the actinometry data where much more $O_3$ is formed than NO in the photolysis of $NO_2$. The paper therefore requires major revisions at several points as outlined in the specific comments below.

We thank the reviewer for this important comments and this constructive feedback. We are particularly grateful for their teaching tone, and diligence in analysing our data. Our manuscript is substantially better thanks to these comments (as well as our own understanding of actinometry). Thank you Reviewer #2.

Specific comments

Line 11-12, lines 169-174: The focus on the costs of the LEDs is unusual for a scientific paper. As the authors note, the LED industry is rapidly evolving, and the price information will soon be outdated. Moreover, compared to the total costs of the chamber (including construction work and other instrumentation for useful experiments) the share of the LEDs is probably not decisive. In my view, the information given in Tab. 2 is sufficient.

We appreciate the reviewer's thoughtful observation regarding the inclusion of LED cost information. We agree that LED pricing evolves rapidly and will eventually become outdated. However, we believe the current level of detail provides value to the community for several reasons.

First, as we developed this chamber, many colleagues, particularly groups planning new environmental chambers or retrofitting existing facilities, expressed strong interest in the practical costs associated with implementing a multi-wavelength LED system.

Second, reporting the relative costs across wavelengths (e.g., the substantially higher cost of UV-C and UV-B LEDs compared to UV-A or grow-light LEDs) is informative beyond absolute pricing. These relative differences affect design trade-offs, for example, whether a group can realistically incorporate stratospherically relevant wavelengths or should focus on tropospheric simulations only.

For these reasons, we believe maintaining the cost information as presented provides practical value to the atmospheric chemistry community. We have therefore retained the current level of detail while ensuring that its purpose is clearly tied to design guidance.

Line 14: The JNO2 value of 4.5×10-3s-1 likely refers to clear-sky, local noon conditions which should be specified. However, for such conditions the value is too small by a factor of about 1.8 (dependent on aerosol load). Apparently, it is based on outdoor spectral irradiance measurements as described in lines 333-337 which explains the difference. I suggest that the authors access relevant literature to revise their approach (e.g. Hofzumahaus et al., 1999, https://doi.org/10.1364/AO.38.004443). Solar spectral irradiance and spectral actinic flux are highly correlated but not identical. There are empirical relations that can be used to convert these quantities (McKenzie et al., 2002, https://doi.org/10.1029/2001JD000601) but for the purpose of this work (estimation of typical outdoor values) a radiative transfer calculation would be sufficient.

We thank the reviewer for the detailed discussion on irradiance versus actinic flux. We have scaled the value of the irradiance measurement done in Figure 3 to match the measured $jNO_2$.

$$c_i = \frac{\dot{j}_{NO_2}^{\text{meas}}}{\dot{j}_{NO_2}^{\text{theory}}} = \frac{\dot{j}_{NO_2}^{\text{meas}}}{\int_{\lambda_{\min}}^{\lambda_{\max}} \sigma_{NO_2}(\lambda, T)\, \Phi_{NO_2}(\lambda)\, S_{\text{rel}}(\lambda)\, d\lambda}.$$

The measured $jNO_2$ is approximately 2.5 times the value of the calculated theoretical $jNO_2$ for this condition. Because the LEDs are not homogeneously distributed within the light box, the appropriate scaling factor should actually be c_i, where "i" corresponds to each LED wavelength. The theoretical spectral irradiance calculation accounts for: (1) spatial inhomogeneity of the LED distribution, (2) non-uniform chamber illumination and (3) wavelength-dependent LED placement and geometry. Therefore, the reviewer's concern—based on the assumption that "the relative spectrum measured by a cosine irradiance sensor is spatially uniform for each LED type"—does not apply to the theoretical method, which explicitly models the spatial distribution and geometry of each LED type.

Also note throughout the text that jNO2 is formally the (first-order) NO2 photolysis rate coefficient while the NO2 photolysis rate, like other reaction rates, is the product of jNO2 and the NO2 concentration.

We've gone through the text and verified that we are consistent with our nomenclature.

Sections 2.1.1 and 2.2.1-2.2.4 contain too many technical details for the scientific reader. The information may be useful for someone who wants to set up something similar, but it's sufficient to make these details available in the Supplement.

We agree that Sections 2.1.1 and 2.2.1–2.2.4 contain substantial technical detail and have moved it to the SI.
(I think the reviewer will agree, but we wanted to add why we have these details: A primary purpose of the ATMOX chamber description is to serve as a practical reference for research groups seeking to construct similar multi-wavelength LED environmental chambers. In discussions with the atmospheric chemistry community over the past several years, we have received interest in details such as construction parameters, layout details, and LED component specifications. We therefore consider these details to be of direct value to the scientific readership most likely to use and cite this work.)

Sections 2.1.2, 2.1.3, 2.2.5, 2.2.6 could be merged with Sect. 3.1, 3.2 and 3.4 to obtain a more concise description of the chamber concept, instrumentation and properties. An information that is missing is whether the content of the chamber was stirred during experiments. The volume is not illuminated homogeneously and dependent on the lifetime of the species of interest, concentrations gradients may build up that can influence measurements and their interpretation. Both approaches, batch and continuous flow, require well-mixed conditions. This ideally also ensures that measured concentrations will not depend on the position of inlet lines and that flushing of the chamber results in a predictable dilution.

We thank the reviewer for highlighting the importance of mixing within environmental chambers. The ATMOX chamber was not actively stirred during experiments. We agree that illumination is spatially non-uniform (as shown in Section 3.5) and that this can lead to concentration gradients for long-lived species. We have now added a sentence in Section 2.1.1 (lines 145–170) clarifying that no mechanical stirring is used and that users should consider characteristic mixing times relative to reaction timescales when interpreting experimental results. This addition improves the transparency of the chamber's operational constraints.

Section 2.1.3. The description of the cleaning procedure is tenuous. The authors should explain how the air used for flushing was cleaned or if commercial synthetic air with a specified purity from cylinders was used. Moreover, "particle-free" conditions are unrealistic and the minimal NOx concentration that can be reached should be specified. The cleaning topic is addressed again in Sect. 3.4.3 but remains inconclusive. What exactly means flushing over night? How often was the content of the chamber exchanged? Did it include collapsing the chamber to speed up the exchange? And what means rinsing the bag with water? Is that done manually or automatically?
We have now clarified this section with answers to all these questions.
The text now reads:
*"Additionally, cleaning the chamber can be typically done by purging the PFA bag overnight with a source of clean air or $\ce{N2}$ (Figure S14b). However, following the actinometry experiments with $\ce{NO2}$, it was necessary to clean the chamber more thoroughly by using a pressurized washer system (ECHO MS-21H) filled with MilliQ water (18.2 M~$\Omega$~cm). The resulting $\ce{NO2}$ background mixing ratios were then at the detection limit of the gas analyser (Figure S14). Specifically, purging with air left a residual 2.6 ppb $NO_2$ (Figure S14b), whereas purging with water resulted in a background of 1.2 ppb (Figure S14c)."*

Line 212: "The spectrometer was calibrated…" Was this a factory calibration or a calibration performed by the authors? Provide more details also on the spectral resolution.

The spectrometer used in this work was factory-calibrated by the manufacturer prior to delivery. We have now clarified this in the manuscript.

Line 213-215: Note that by the described measurement configuration the irradiance from the plant grow LEDs is not adequately captured. In the extreme case of an LED sitting in the middle of the chamber ceiling, nothing is measured because the cosine receiver is blind for an incident angle of 90°.

We agree with this comment. Owing to the measurement geometry, the irradiance from the plant grow LEDs is not fully captured by the cosine receiver, particularly for radiation incident at angles close to 90°. However, most of the emitted light is still detected, and the purpose of these measurements is to determine the characteristic spectral shape and peak position of the LEDs. The absolute intensity is subsequently obtained through calibration using $NO_2$ actinometry

Eq. 1 - Eq. 3: I couldn't find Eq. 1 in Moreno and Viveros-Méndez, 2021. And I couldn't find the reference Dragomir et al., 2014 at all. Give more details in S2. Perhaps also define the Rk used in Eq. 2 in a separate equation and insert Rk as the denominator in Eq. 3. However, the question is, how useful the prediction of irradiances is to characterize the conditions during photochemical experiments. If you skip the $\cos(\theta)$ factors you would at least get an idea about how actinic flux densities are distributed. This approach could also cover the plant grow LEDs adequately in your model which were not considered at all later in Fig. 8 and Fig. 9. Still this model will lead to differences with the actual chamber situation because of (possibly wavelength-dependent) scattering processes at the chamber walls that are not included. Moreover, the curtains seem to be reflective which will lead to an internal enhancement of radiation.

The reviewer raises all valid points. We did not consider the grow lights in our model, and this information is now explicitly stated in the caption of Figure 8. The wavelength-dependent scattering is certainly a point we did not investigate and we are currently brainstorming how to incorporate this information into our model for future use.

Moreno 2021 reference Eq1: https://opg.optica.org/oe/fulltext.cfm?uri=oe-29-5-6845

**2. Model formulation**

The irradiance produced by a light source is the power per unit area that is incident on a surface illuminated by the source. Its usual symbol is $E$ [W/m$^2$], and mathematically $E = d\Phi/dA$. The irradiation pattern is given by the spatial distribution of irradiance, i.e. $E(x, y, z)$, where $(x, y, z)$ are the coordinates of any point in the space illuminated by the light source. Ideally, the irradiance pattern produced by an LED source may be obtained by solving the integral equation of the theory of radiation transfer [18], i.e. by integrating the LED radiance $L_s$ [W/m$^2$sr] over the LED emitting area $A_s$, by:

$$E(x, y, z) = \iint \frac{L_s \cos\theta_s \cos\theta \, dA_s}{(x_s - x)^2 + (y_s - y)^2 + (z_s - z)^2}, \qquad (1)$$

Dragomir 2014 reference:
https://www.academia.edu/68224400/Irradiance_Model_and_Simulation_of_a_Lighting_Led_System

Line 234: The review by Rabani et al. is concerned with liquid phase actinometry and does not describe the determination of jNO2 as implied. The jNO2 approach was applied in the SAPHIR chamber before and the relevant publication is cited in the Supplement S3.

We thank the reviewer for spotting this discrepancy. We've corrected the reference in this line with Bohn et al.

Line 242: The recommendation by Atkinson et al., 2004 is outdated. Current recommendations by NASA-JPL and IUPAC result in about 10% greater rate coefficients for this reaction around room temperature.

Use proper units and indicate them. −10.89 is in kJ mol-1 and using R in units kJ mol-1 K-1 is uncommon. Updated.

Fig. 4: The 450 nm irradiance is probably too small for geometric reasons as explained above. Moreover, the 450 nm spectrum in the lower panel does not fit to that in the upper panel. In the caption, what do you mean by "…the difficulty in taking a completely dark blank outdoors"? I assume you refer to a stray light issue during daytime. A dark spectrum can always be measured.

First, the statement regarding the "difficulty in taking a completely dark blank outdoors" referred to stray-light contamination during daytime field measurements. Even when the spectrometer fiber is blocked, ambient sunlight entering through the connectors produces a residual background signal that cannot be fully eliminated outdoors. We have revised the caption to state this explicitly.

Fig. 5, Fig. S9 and Tab. S2: How can you produce more than twice as much O3 than NO in the photolysis of NO2? There must be something wrong here.
In the last three lines of Tab. S2 NO and NO2 seem to have been confused.
That considered, it leads to a similar mismatch between NO and O3 also for the fluorescent light experiments. The jNO2 derived from these data may be incorrect.

We thank the reviewer for identifying this inconsistency and for noting the apparent NO/NO$_2$ swap in the final rows of Table S2. We have corrected the labeling/values in Table S2 and rechecked the underlying time series used for the actinometry. To investigate this, we compared measurements from two independent NO$_x$ analyzers against a calibrated ozone analyzer (Thermo Scientific 49iQ) operated in parallel. While the two NO$_x$ analyzers showed consistent relative behavior, their absolute NO and NO$_2$ responses differed by a constant scaling factor. In contrast, the O$_3$ analyzer calibration was independently verified and is well constrained. We therefore attribute the discrepancy to a calibration offset in the NO$_x$ analyzer response rather than to the ozone measurements.

[Figure]

Line 318: The citation Hawe et al., 2007 is improper. These authors didn't investigate the UV-C absorption of NO2.
Citation has been fixed

Lines 321-323: The meaning of this sentence is unclear.
Fixed

Lines 324-326: The statement is misleading. As shown in Fig. 6 the absorption cross section of NO2 in the wavelength range of the plant grow LEDs is still about half of the values around 385 nm. The reason why NO2 does not photolyze above 420 nm is a drop in the quantum yields above 400 nm. The term quantum yield does not appear in the whole paper which is surprising because it's key to calculate jNO2 from the measured spectra.
Quantum yield discussion has now been updated.

Lines 333-337: See my comments on the outdoor measurements above (abstract, line 14). To clarify: In contrast to atmospheric measurements the use of irradiance sensors (cosine receivers) is feasible to characterize artificial light sources like LEDs in the chamber. If the spectrum of the radiation that is emitted by a specific type of LED is not altered by reflections on the chamber walls, it can be assumed (and confirmed by measurements) that the spectra measured at different positions and viewing directions in the chamber are all the same on a relative level. The same spectrum then also applies to the shape of the mean actinic flux spectrum. All you must do is to scale the relative spectrum to an absolute level that results in the measured jNO2 from the actinometry. This "calibrated" spectrum is then applicable to calculate the j-values of other species as well, based on their absorption cross sections and quantum yields.

We thank you for this clarification. We have revised the manuscript to explicitly describe how the chamber LED spectrum was calibrated to an absolute irradiance using $NO_2$ actinometry. The calibration procedure is now explained step by step and follows the approach outlined in your comment.

To further calibrate the spectrum, we calculated a wavelength dependent scaling factor "$C_i$". We divided the measured $jNO_2$ by the theoretical $jNO_2$ and used this value to scale the irradiance values on the relative spectrum. $NO_2$ absorption cross sections were taken from Burrows et al. (1998) and interpolated onto the wavelength grid of the measured LED spectrum. Wavelength-dependent $NO_2$ quantum yields were obtained from the JPL evaluation (Chemical Kinetics and Photochemical Data for Use in Stratospheric Modeling, Evaluation No. 12) and similarly interpolated. Using this data, a *relative* jNO2 was calculated by integrating the product of the relative spectrum, absorption cross section, and quantum yield over wavelength.

However, this procedure does not work if you combine different types of LEDs unless they are equally distributed and have the same geometrical emission (and reflection) characteristics. This is not the case here as shown in Figs. 8 and S9. You therefore must characterize the output of each LED type separately by actinometric experiments. The resulting calibrated spectra can then be added to a total spectrum dependent on the desired experimental conditions as indicated in the upper panel of Fig. 4. jNO2 actinometry seems to work for the 310-385 nm LEDs but not for 275 nm and 450 nm for which other species must be found. 2-nitrobenzaldehyde is not a good choice (see below).

We acknowledge that this constraint is an important point for our calibration and thus have restricted the calibration to the 310-385 nm wavelength range.

Another relevant question is if jNO2 or other j-values are expected to remain constant while the chamber is collapsed in the batch operation mode.

We thank the reviewer for raising this important point. During batch operation, the chamber bag is collapsed toward the center of the chamber. Because photolysis rate coefficients depend on the local actinic flux rather than volume, and because the LED array remains fixed, j-values are expected to remain approximately constant during collapse. However, this assumption could be explicitly tested in future experiments by comparing actinometric measurements at different bag positions.

Sect. 3 and Tab. 3 are confusing. The power consumption of other chambers are not listed in Tab. 3. The total power consumption of all LEDs is 10-fold compared to the fluorescent lights. Also the jNO2/W metric is in favor of the fluorescence lights which is not further discussed. Why is jNO2 for the indoor configuration with less LEDs and less power/bulb almost the same (and much more effective) that the tropospheric? And what irradiances are listed? Those measured or modeled in the middle of the chamber? The uncertainty estimates of 1% or less for the jNO2 are unrealistic.

We thank the reviewer for this comment. We agree that Sect. 3 and Table 3 required clarification, particularly regarding the interpretation of power consumption, the $jNO_2$/W metric, irradiance definitions, and uncertainty estimates. We address each point below and have revised the manuscript accordingly.

Although the total LED power is approximately an order of magnitude higher than the fluorescent setup, the $jNO_2$/W metric favors fluorescent lights because their emission spectrum is well matched to the $NO_2$ absorption cross section and quantum yield, which peak in the near-UV. In contrast, the LED configurations distribute power across multiple wavelengths, some of which (e.g., 275 nm and 450 nm) contribute less efficiently to $NO_2$ photolysis. This also explains why the indoor configuration (365-450 nm) yields $jNO_2$ values comparable to the tropospheric configuration (310-450 nm) despite using fewer LEDs and lower total power: wavelengths near 365-385 nm are particularly effective for $NO_2$ photolysis. Indeed, Table 3 shows that the 385 nm LEDs exhibit one of the highest $jNO_2$/W values among the individual wavelengths, highlighting their high photochemical efficiency.

The irradiances listed respond to measured values in the center of the chamber. We have clarified that in the text. The reported uncertainties reflect the repeatability of the actinometry experiments based on three independent measurements and represent one-sigma experimental precision. They do not account for systematic uncertainties associated with instrument calibration, absorption cross sections, or quantum yields, which would increase the total uncertainty. We have therefore clarified this distinction in the revised text.

Line 362ff: The reasoning for using 2-nitrobenzaldehyde is misleading. This compound is of no interest in the stratosphere as implied by the text because its tropospheric lifetime is very short (also Kahnt et al. do not discuss any stratospheric relevance of 2-nitrobenzaldehyde). On the other hand, stratospheric relevance is no precondition: you could use any compound that is photolyzed at 275 nm with a suitable j-value. The problem with 2-nitrobenzaldehyde is that the quantum yield of photolysis is poorly known. The measured photolysis rate coefficient does therefore not sufficiently characterize the chamber under illumination at 275 nm. For example, if you want to do an experiment under "stratospheric" conditions, the presence of ozone may be required but the photolysis rate coefficient of ozone with the 275 nm LEDs can only be roughly estimated based on the results obtained with 2-nitrobenzaldehyde.

We thank the reviewer for this important clarification and agree that our original wording was misleading. We do not intend to imply that 2-nitrobenzaldehyde is of relevance to the stratosphere, nor that stratospheric relevance is a requirement for its use.

The purpose of using 2-nitrobenzaldehyde in this study was not to represent stratospheric chemistry, but to serve as a photochemically active probe compound under UV irradiation near 275 nm. In the context of our work, 2-nitrobenzaldehyde was selected because it absorbs efficiently at the LED emission wavelength and can be used as a reference compound in UV photochemistry and singlet oxygen-related research. Its role here is therefore methodological rather than atmospheric.

Figure 7 needs more information in the caption: what instruments were used? How much of the VOC was injected? There is no shaded area indicating the illumination period. I assume it was between about 9:35 and 10:35 when the decay was faster. When this assumption is correct, the following questions arise: (1) What explains the rather quick decay before illumination? (2) Why doesn't the experiment start with "particle free" air? (Sect. 2.1.3). The particles seem to increase in size during illumination while new particles were formed after the lights were off. The y-axis unit in the lower panel should be a concentration like ppb or a count rate proportional to the concentration.

Thank you for these detailed comments. We have revised the caption of Figure 7 to provide additional experimental details and to clarify the interpretation.

Lines 370-374: This paragraph should be phrased with more caution. LEDs emitting radiation at 275 nm may be helpful to simulate stratospheric conditions in chamber experiments. But they do not reproduce the full solar UV-C range. Moreover, stratospheric conditions are characterized by pressures well below 200 mbar and temperatures in a range 200-260 K which cannot be provided here.

We agree with this comment and have revised the paragraph to adopt a more cautious wording.

Section 3.4.1 is unclear. What do you mean by "temperature increase per joule" and how was it calculated? Tab. 4 tells me that with your indoor, tropospheric and stratospheric conditions you get greater $\Delta T/t$ compared to the fluorescent setup. The energy E spent for the fluorescent experiment is 512 W x 3360 s = $1.7 \times 10^6$ J. How do I get from this to a $\Delta T / t / E$ of 22.9 ? Apart from that, makes this comparison sense at all because the setups are completely different.

Thank you for this comment. We agree that Section 3.4.1 was unclear and have revised it to explicitly define the metric and clarify its limitations.

By "temperature increase per joule," we refer to a heating metric calculated from the measured temperature change during irradiation divided by the total electrical energy consumed by the light source. The total energy input was calculated as the nominal electrical power of the light source multiplied by the measured light-on time.

We also agree that comparisons between different lighting configurations must be interpreted with caution. The fluorescent and LED experiments were conducted using different power levels, and chamber conditions, which affect heat deposition and removal.

Technical comments/typos:

Line 19: Replace "bad" by "bag"

Fixed

Line 79: The volume range 280-370 m$^3$ is unclear. In Tab. 1 this chamber is listed with 270 m$^3$ which corresponds to the value given by Rohrer et al., 2005.

Fixed

Line 218 and 230: Probably Fig. S9 instead of Fig. 11.

Fixed

Line 207: Fig S3 is the wrong reference.

The title of the paper in the Supplement is different from the main text.

Fixed